# Leveraging genetic diversity to identify small molecules that reverse mouse skeletal muscle insulin resistance

Stewart WC Masson[1], Søren Madsen[1], Kristen C Cooke[1], Meg Potter[1], Alexis Diaz Vegas[1], Luke Carroll[2], Senthil Thillainadesan[1], Harry B Cutler[1], Ken R Walder[3], Gregory J Cooney[1], Grant Morahan[4], Jacqueline Stöckli[1], David E James[1,5]*

[1]Charles Perkins Centre, School of Life and Environmental Sciences, University of Sydney, Camperdown, Australia; [2]Australian Proteome Analysis Facility, Macquarie University, Macquarie Park, Australia; [3]School of Medicine, Deakin University, Geelong, Australia; [4]Centre for Diabetes Research, Harry Perkins Institute of Medical Research, Murdoch, Australia; [5]School of Medical Sciences University of Sydney, Sydney, Australia

*For correspondence:
david.james@sydney.edu.au

**Abstract** Systems genetics has begun to tackle the complexity of insulin resistance by capitalising on computational advances to study high-diversity populations. 'Diversity Outbred in Australia (DOz)' is a population of genetically unique mice with profound metabolic heterogeneity. We leveraged this variance to explore skeletal muscle's contribution to whole-body insulin action through metabolic phenotyping and skeletal muscle proteomics of 215 DOz mice. Linear modelling identified 553 proteins that associated with whole-body insulin sensitivity (Matsuda Index) including regulators of endocytosis and muscle proteostasis. To enrich for causality, we refined this network by focusing on negatively associated, genetically regulated proteins, resulting in a 76-protein fingerprint of insulin resistance. We sought to perturb this network and restore insulin action with small molecules by integrating the Broad Institute Connectivity Map platform and in vitro assays of insulin action using the Prestwick chemical library. These complementary approaches identified the antibiotic thiostrepton as an insulin resistance reversal agent. Subsequent validation in ex vivo insulin-resistant mouse muscle and palmitate-induced insulin-resistant myotubes demonstrated potent insulin action restoration, potentially via upregulation of glycolysis. This work demonstrates the value of a drug-centric framework to validate systems-level analysis by identifying potential therapeutics for insulin resistance.

## eLife assessment

This **fundamental** study leverages natural genetic diversity in mice to discover candidate genes for insulin sensitivity, followed by experimental identification of compounds that can modulate insulin sensitivity, and finally initial mechanistic investigation of the mode of action. The generalized approach presented here, - the integration of systems genetics data with drug discovery -, supported by **compelling** evidence, will be an **important** guide for others that seek to translate insights from mammalian genetics to drug discovery.

## Introduction

Skeletal muscle is a key determinant of whole-body glycaemic control. Under optimal conditions, insulin secreted from the pancreas initiates a signalling program in muscle and other tissues, culminating in translocation of the insulin-responsive glucose transporter (GLUT4) to the plasma membrane

(*Bryant et al., 2002*). Increased plasma membrane GLUT4 lowers circulating glucose by increasing cellular influx for either storage as glycogen or subsequent metabolism via the glycolytic pathway. Insulin resistance is the progressive failure of these processes and often precedes a number of metabolic disorders, including type 2 diabetes (*James et al., 2021*). Advances in genomics and computational biology have begun to shed new light on molecular drivers of insulin resistance. A recent study in humans undertook a GWAS of 188,577 individuals unveiling 53 genetic loci associated with a surrogate insulin resistance signature (*Lotta et al., 2017*). Such studies are important as they point towards genetic lesions in metabolic tissues like skeletal muscle and adipose tissue as playing a key causal role in the development of insulin resistance. This emphasises the importance of focusing on genetics and peripheral tissues for new therapeutic targets and strategies to overcome insulin resistance. Recent developments in systems biology provides unique opportunities for discovering ways of reversing and/or preventing insulin resistance. This will have enormous clinical benefits since insulin resistance is a gateway to an expanding family of diseases.

Over the past decade, genetically diverse mouse panels have been used to study metabolic diseases (*Nelson et al., 2022*; *Parks et al., 2015*; *Williams et al., 2016*; *Yang et al., 2020*). This is a major step forward because these panels combine control of the environment and access to any biological tissue, with a vast phenotypic and range which can be leveraged towards understanding complex diseases. Three specific resources are the hybrid mouse diversity panel (HMDP), and the BXD (C57BL/6J × DBA) and Collaborative Cross (CC) mouse strains (*Yang et al., 2020*; *Ghazalpour et al., 2012*; *Ashbrook et al., 2021*; *Peirce et al., 2004*; *Collaborative Cross Consortium, 2012*). These panels comprise large selections of inbred mice spanning vast phenotypic and genetic landscapes. CC mice were first generated by interbreeding five commonly used laboratory mouse strains (C57BL/6J, A/J, 129S1/SvlmJ, NZO/HILtJ, NOD/ShiLtJ) and three wild-derived strains (WSB/EiJ, CAST/EiJ, PWK/PhJ) in a 'funnel' design. The resulting CC strains were then outbred to generate Diversity Outbred mice at Jackson Laboratories (*Svenson et al., 2012*) which have increased phenotypic diversity and resolution for genetic mapping. An independent Diversity Outbred colony was established in Western Australia using CC mice from Geniad (*Ferguson et al., 2019*). This colony has since relocated to our group at the University of Sydney, termed Diversity Outbred mice from Australia (Oz) or DOz.

The use of such rodent models for studying complex traits has given rise to the field of systems genetics. System genetics uses global quantification of 'intermediate phenotypes', that is, gene transcripts, proteins, and metabolites, to provide mechanistic links between genetic variation and complex traits/diseases (*Baliga et al., 2017*; *Seldin et al., 2019*). Unlike traditional genetic studies that identify single or multiple loci of interest, systems genetics often identifies entire biological pathways or networks that are inherently more difficult to empirically test. In an attempt to streamline interrogation of molecular pathways, several large-scale perturbation screens/projects have been undertaken. One such example is the Broad Institute's Connectivity Map (CMAP) that integrated mRNA expression levels from 1.5 million combinations of different cell lines and perturbations (small-molecule inhibitors, receptor ligands, genetic manipulations) into a searchable database (*Subramanian et al., 2017*; *Uva et al., 2021*; *Lamb et al., 2006*; *Lamb, 2007*). These kinds of tools provide invaluable resources for testing hypotheses generated from systems genetics experiments and broadly linking molecular networks to phenotypic outcomes.

Here we have utilised DOz mice to interrogate insulin resistance. By combining metabolic phenotyping and skeletal muscle proteomics, we have identified an insulin resistance fingerprint of 76 proteins. We then used CMAP to identify small molecules that give rise to an overlapping transcriptional signature across a number of cell lines, and therefore have the potential to affect insulin action. Strikingly, one of these compounds, the antibiotic thiostrepton, was also identified by us in an independent small-molecule screen for effectors of insulin action in myotubes. Subsequent validation of thiostrepton uncovered profound beneficial effects on insulin resistance in vitro and ex vivo, potentially via modulation of mitochondrial function and glycolysis.

## Results
### DOz metabolic and proteomic variation
DOz mice were metabolically phenotyped by oral glucose tolerance test (GTT) and echoMRI to determine body composition (*Figure 1A–C*). We integrated glucose and insulin levels during the GTT into

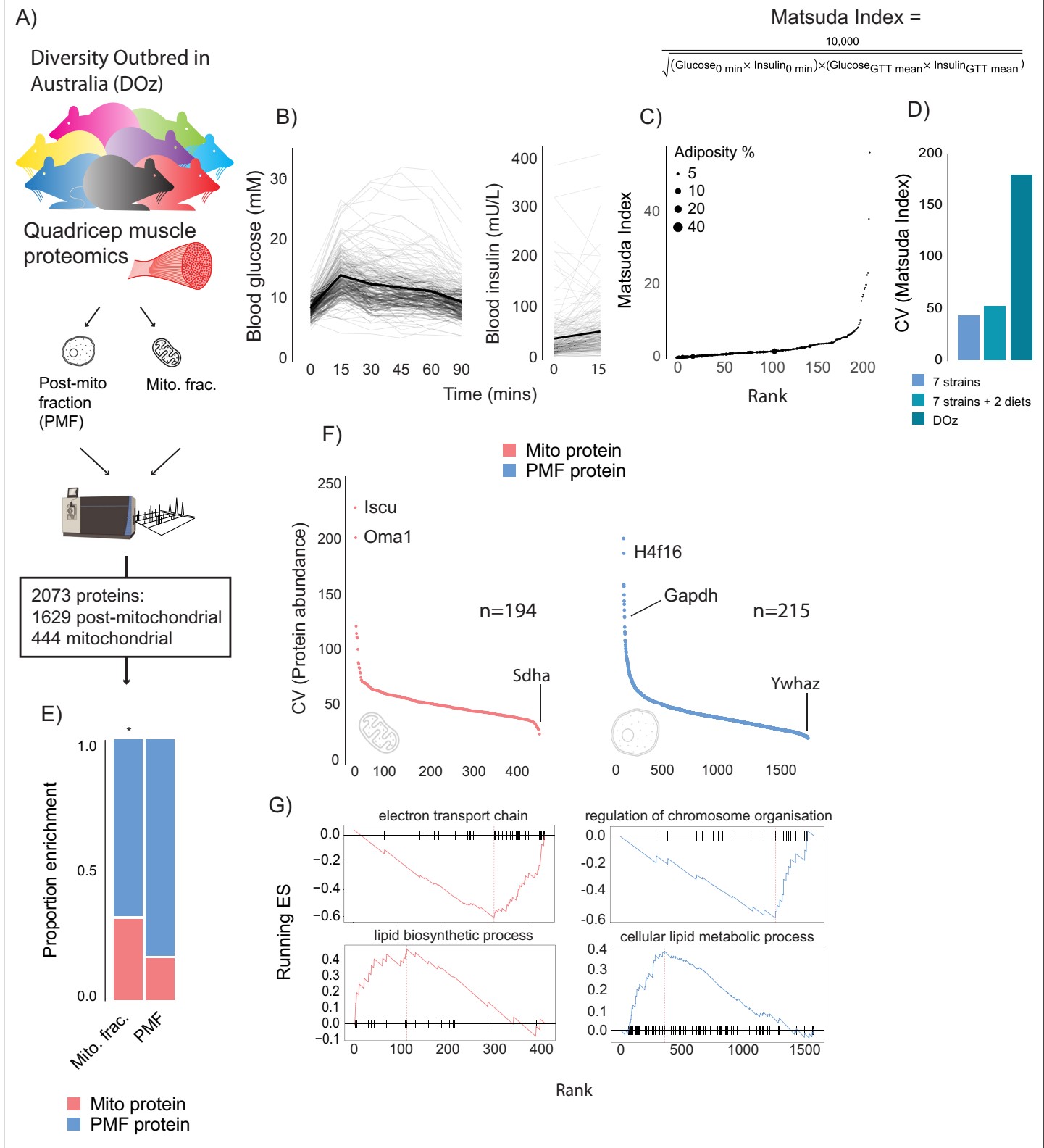

**Figure 1.** Metabolic and proteomic diversity of Diversity Outbred in Oz (DOz) mice. (**A**) Schematic of metabolic phenotyping and quadricep proteomics in chow-fed DOz mice. (**B**) Blood glucose and insulin levels during a glucose tolerance test (GTT). (**C**) Whole-body insulin sensitivity (Matsuda Index, formula shown above) and adiposity of DOz mice (n = 215). (**D**) Comparison of coefficient of variation (CV) of insulin of Matsuda Index across inbred strains and diets versus chow-fed DOz mice. (**E**) Relative enrichment of mitochondrial (Mito) proteins in mitochondrial fraction and post-mitochondrial

*Figure 1 continued on next page*

*Figure 1 continued*

fraction (PMF) of quadricep proteomes. (**F**) Relative protein CV across mitochondrial and post-mitochondrial quadricep fractions. (**G**) Biological pathways enriched in mitochondrial and post-mitochondrial quadricep fractions, running enrichment score for a given pathway (ES) on y-axis and proteins ranked by CV on x-axis. Significance testing was performed by chi-square test. * indicates a significant difference p<0.01.

The online version of this article includes the following figure supplement(s) for figure 1:

**Figure supplement 1.** Assessment of variation-based enrichment analysis Diversity Outbred in Oz (DOz) muscle proteomics.

a surrogate measure of whole-body insulin sensitivity referred to as the Matsuda Index (*Matsuda and DeFronzo, 1999*). Similar to HOMA-IR, the Matsuda Index uses blood glucose and insulin values to predict whole-body insulin sensitivity. However, an advantage of the Matsuda Index over HOMA-IR is that it includes values over a range of GTT timepoints, which better incorporates the dynamics of glycaemic control. Furthermore, the Matsuda Index is better correlated to the euglycemic-hyperinsulinaemic clamp, the gold standard measurement for insulin sensitivity, in humans (*Matsuda and DeFronzo, 1999*). Consistent with studies in other DO mouse populations (*Svenson et al., 2012*; *Churchill et al., 2012*), we observed profound phenotypic diversity in DOz mice with 20- to 400-fold differences in insulin sensitivity, adiposity, and fasting insulin levels across all DOz animals (*Figure 1C*). Notably, the metabolic variation we observed in DOz mice is markedly greater than the variation typically observed in similar studies using inbred mouse strains (*Figure 1D*). Since skeletal muscle and mitochondrial function are major contributors to whole-body insulin sensitivity in mammals (*Nelson et al., 2022*; *DeFronzo, 1987*; *Anderson et al., 2009*), we performed proteomic analysis on quadriceps muscles that were fractionated into mitochondrial and post-mitochondrial fractions (PMF; *Figure 1A*). We identified a total of 2073 proteins (444 mitochondrial and 1629 PMF) present in at least 50% of mice. Mitochondrial proteins were defined based on their presence in MitoCarta 3.0 (*Rath et al., 2021*) and consistent with previous work (*Williams et al., 2018*) were approximately twofold enriched in the mitochondrial fraction relative to the PMF (*Figure 1E*).

As with glycaemic control, muscle proteomes exhibited profound variation, with approximately 200-fold differences in coefficient of variation (CV) across both fractions (*Figure 1F*). Interestingly, glyceraldehyde-3-phosphate dehydrogenase (GAPDH), the often-reported western blot loading control, was the 11th most variable protein in our dataset. Amongst the other highly variable proteins were the histone subunit H4F16, the mitochondrial iron-sulphur cluster assembly regulator ISCU, and the metalloendopeptidase OMA1. Among proteins with low variability between mice was the mitochondrial respiratory complex II subunit SDHA and the 14-3-3 zeta isoform YWHAZ. To uncover how variation differed across biological processes, Gene Ontology (GO) enrichment analysis was performed on proteins ranked by CV. Among the low-variance processes were the electron transport chain (mitochondrial fraction) and regulation of chromosome organisation (PMF), while lipid metabolic pathways were highly variable across both mitochondrial and post-mitochondrial fractions (*Figure 1G*). As a control experiment, we also performed enrichment analysis on proteins ranked by LFQ relative abundance. High CV pathways (enriched for high CV proteins) tended to be lower in relative abundance (enriched for low relative abundance proteins) (*Figure 1—figure supplement 1A and B*). However, many high-variability pathways, lipid metabolism, for example, were not enriched in either direction based on relative abundance suggesting that differences in relative abundance do not fully explain pathway variability differences.

## Role of skeletal muscle in whole-body insulin sensitivity

To leverage genetic and metabolic diversity towards uncovering new regulators of insulin action, we constructed linear models comparing insulin sensitivity (Matsuda Index) against protein abundance. Our initial analysis identified 37 mitochondrial and 40 PMF proteins that significantly associated with the Matsuda Index. Many of these appeared to be involved with adiposity rather than insulin action including adiponectin (ADIPOQ), adipsin (CFD), and the mitochondrial β-oxidation proteins ETFA and ETFB (*Figure 2—figure supplement 1A and B*). Because we were mainly interested in identifying muscle-specific factors that regulate insulin sensitivity, we next constructed a model that included adiposity (the percentage of body mass which is adipose tissue) as a covariate. Using this approach, we identified 120 mitochondrial (*Figure 2A and B*) and 433 PMF proteins that significantly associated with whole-body insulin sensitivity. A comparison of the results from each model (with and without adiposity as a covariate) revealed consensus proteins that associated with insulin sensitivity

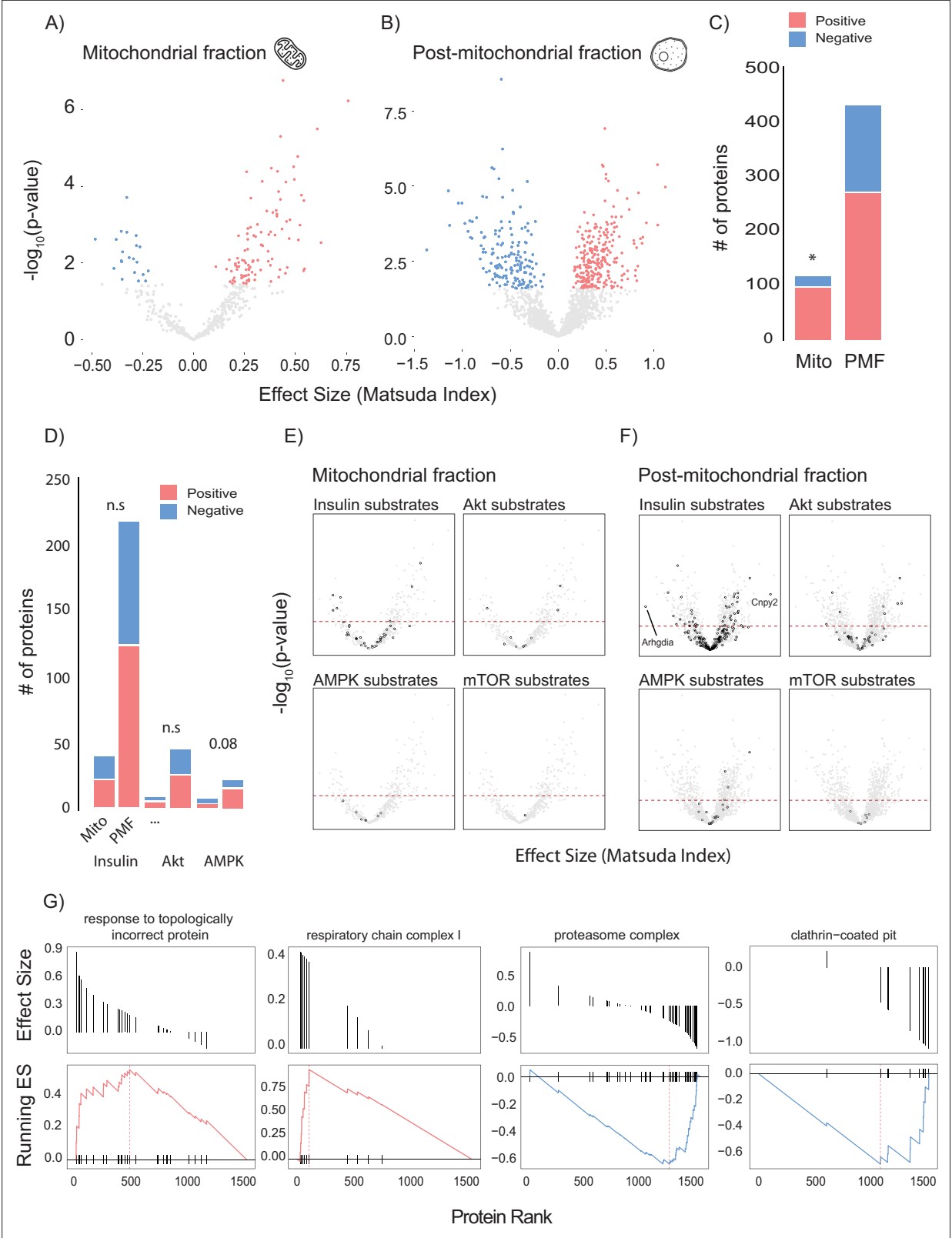

**Figure 2.** Linear modelling of quadricep proteome and whole-body insulin sensitivity. (**A, B**) Volcano plot with Matsuda Index effect sizes (x-axis) and significance (y-axis) for mitochondrial (**A**) and post-mitochondrial (**B**) quadricep proteins using a linear model with adiposity as a covariate. Significant proteins with positive and negative effect sizes are indicated in red and blue, respectively. (**C**) Comparison of positively and negatively associated proteins between fractions. (**D**) Number of proteins identified in each fraction with known roles in insulin or AMPK signalling. (**E, F**) Volcano plot shown

*Figure 2 continued on next page*

*Figure 2 continued*

in (**A**) with mitochondrial (**E**) and post-mitochondrial fraction (**F**) proteins shown in black that have documented roles in indicated signalling pathways. Adjusted p-value threshold is indicated (red dotted line). (**G**) Pathways enriched for proteins which positively and negative associate with Matsuda Index. Effect sizes of proteins within a given pathway and running enrichment score (ES) for a that pathway on y-axis, and proteins ranked by CV on x-axis. Proteins ranked by Matsuda Index effect size on x-axis. Linear modelling was performed using a Gaussian distribution with q-value adjustment of p-values. Enrichment tests between fractions were performed by chi-square test. * indicates a significant difference p<0.01.

The online version of this article includes the following figure supplement(s) for figure 2:

**Figure supplement 1.** Comparison of linear modelling approaches for insulin sensitivity and muscle proteomics.

independently of model design (*Figure 2—figure supplement 1C*). These included the glycolytic enzymes PFKFB1 and PKM, which have previously been identified as regulating muscle insulin action (*Nelson et al., 2022*). Using adiposity as a covariate not only increased the number of proteins identified but also uncovered relationships between Matsuda Index and the GLUT4 trafficking regulators RAB10 (*Su et al., 2018*; *Chen and Lippincott-Schwartz, 2013*), SNAP23 (*Kawanishi et al., 2000*), and IQGAP1 (*Chawla et al., 2017*) which were not seen in the original model. A comparison of the mitochondrial and PMF proteomes revealed a mitochondrial enrichment for proteins that positively associate with insulin sensitivity, highlighting the broadly positive role mitochondria play in muscle's contribution to metabolic health (*Figure 2C*).

To gain further insight into the biology of skeletal muscle glycaemic control, we annotated candidate proteins with PhosphoSite Plus data to test whether any protein of interest had documented roles in metabolic signalling (*Figure 2D–F*). We observed no relationship between insulin signalling substrates and regulation of whole-body insulin sensitivity (*Figure 2D*). However, there was a trend (p=0.08) for AMPK substrates in the PMF to be positively associated with insulin sensitivity. Interestingly, the most positively (Canopy2; CNYP2) and most negatively (Rho GDP dissociation inhibitor alpha; ARHGDIA) associated proteins were both annotated as being insulin responsive (*Figure 2F*). We also preformed GO gene set enrichment analysis (GSEA) on proteins ranked by Matsuda Index effect sizes (*Figure 2G*). In the PMF, 'response to topologically incorrect protein: GO:0051788' was positively associated with insulin sensitivity while the 'proteasome complex: GO:0005839' and 'clathrin-coated pit: GO:0005905' were negatively associated. Together the unfolded protein response and proteasome are indicative of the role of proteostasis in insulin sensitivity (*Díaz-Ruiz et al., 2015*; *Minard et al., 2016*). Conversely, clathrin-coated pits play an important role in the internalisation of the insulin-sensitive glucose transporter GLUT4 from the cell surface, a process negatively regulated by insulin (*Antonescu et al., 2008*; *Robinson et al., 1992*; *Fazakerley et al., 2010*). Consistent with our observation that the majority of the mitochondrial proteome is positively associated with insulin sensitivity, GSEA did not produce any negative results but did uncover a positive relationship between the mitochondrial respiratory complex I and Matsuda Index.

## Integration of genetic linkage analysis and linear modelling with Connectivity Map

Changes in protein levels may be either cause or consequence of changes in insulin sensitivity. In an attempt to select for proteins with a potentially causal relationship, we performed genetic mapping analysis of both the mitochondrial and PMF proteomes (*Figure 3A*). Across both proteomes, we identified 624 protein quantitative trait loci (pQTL). These were distributed across the genome and were predominately *cis*-acting (*Figure 3A*), indicating that a significant proportion of variation in these proteins can be explained by their local genetic architecture. Next, we filtered proteins that were negatively associated with Matsuda Index by *cis*-pQTL presence to generate a molecular fingerprint of insulin resistance (*Figure 3B*). We focussed on negatively associated proteins based on the assumption that inhibiting deleterious proteins is easier than promoting the activity of beneficial ones. Filtering based on *cis*-pQTL presence was based on the rationale that if genetic variation can explain protein abundance differences between mice, then we can be confident that phenotype (Matsuda Index) is not driving the observed differences and therefore the protein-to-phenotype associations are likely causal. Importantly, this assumption can only be made for *cis*-acting pQTLs. Our analysis yielded a list of 76 (69 PMF and 7 mitochondrial) proteins that encompassed a wide range of biological processes (*Supplementary file 1*). Low mitochondrial representation in the fingerprint is the result of selecting negatively associating proteins, and as seen (*Figure 2C*) previously, the mitochondrial

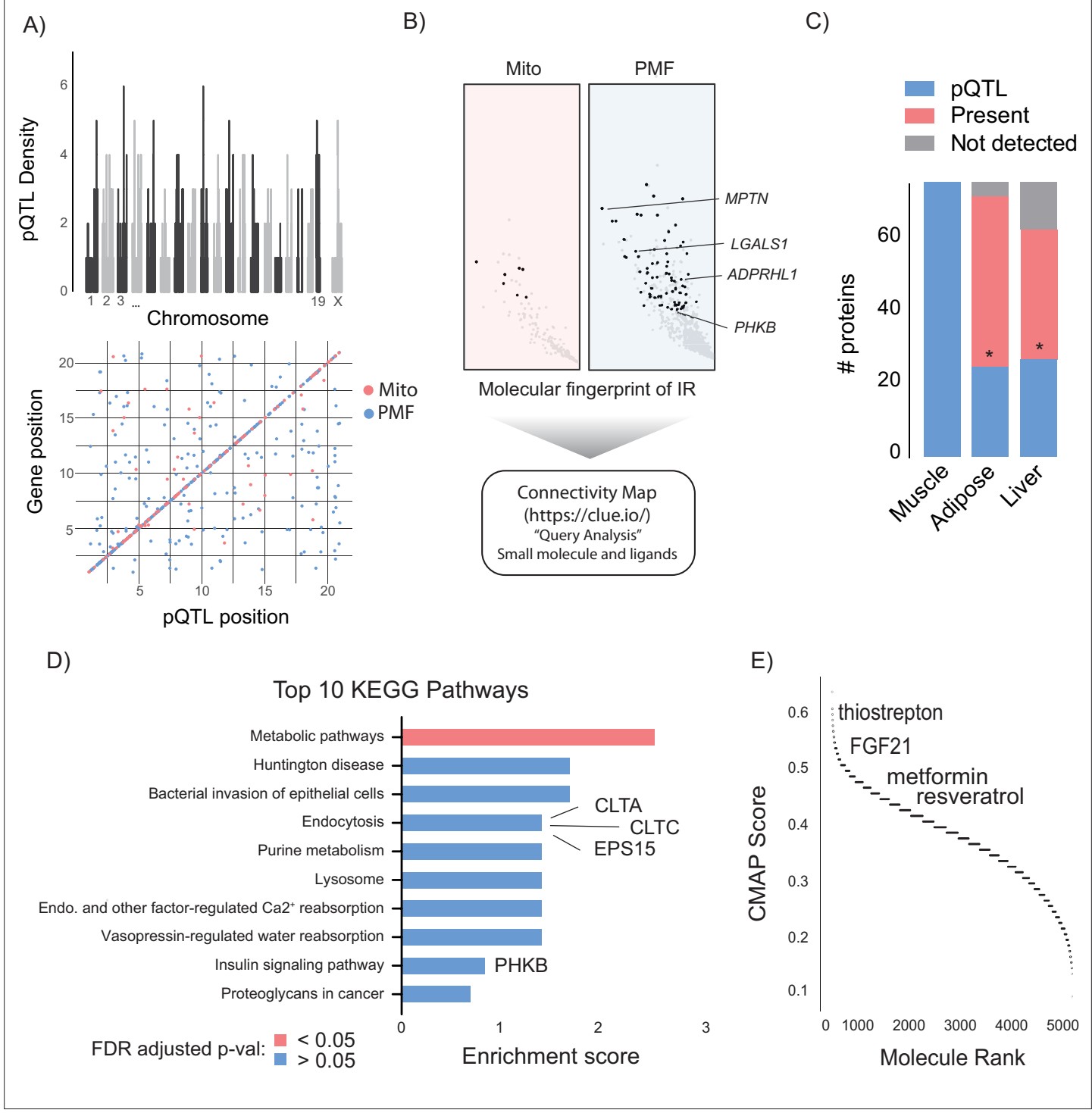

**Figure 3.** Integration of proteomic data via Connectivity Map (CMAP). (**A, B**) Workflow includes filtering for proteins with *cis*-pQTL (**A**) and negative association with Matsuda Index (**B**) prior to CMAP query. (**A**) Distribution of *cis*- and *trans*-pQTL across mitochondrial and post-mitochondrial fraction (PMF) proteome. (**B**) Left side of volcano plots from *Figure 2A and B* (proteins negatively associated with Matsuda Index) is shown with proteins comprised in molecular fingerprint of insulin resistance (IR) indicated in black. Proteins with human homologues are highlighted. (**C**) Comparison of molecular fingerprint of insulin resistance across muscle adipose and liver proteomes. (**D**) Top 10 KEGG pathways enriched in the molecular fingerprint of insulin resistance with proteins of interest highlighted. (**E**) Distribution of CMAP scores for identified small molecules and ligands with compounds of interest indicated. Significance testing was performed by chi-square test. * indicates a significant difference p<0.001. pQTL, protein quantitative trait loci.

*Figure 3 continued on next page*

*Figure 3 continued*

The online version of this article includes the following figure supplement(s) for figure 3:

**Figure supplement 1.** Assessment of protein quantitative trait loci (pQTL) filtration as a method to improve Connectivity Map (CMAP) compound identification.

proteome is enriched for positive contributors to insulin resistance. Similar approaches to identify molecular phenotypes of insulin resistance have previously been conducted using collated human transcriptomic datasets (*Sears et al., 2009*; *Gallagher et al., 2010*; *Leng et al., 2010*; *Josse et al., 2011*; *Phillips et al., 2013*; *Sood et al., 2015*; *Barberio et al., 2016*; *Espah Borujeni et al., 2016*; *Hangelbroek et al., 2016*; *Nakhuda et al., 2016*; *Phillips et al., 2017*). Using a compiled list from *Timmons et al., 2018*, we searched for orthologues of proteins from our fingerprint that associate with human insulin resistance. We identified four such genes (*Figure 3B*): *MTPN* (myotrophin), *LGALS1* (galectin 1), *PHKB* (phosphorylase b kinase), and *ADPRHL1* (ADP-ribosylhydrolase Like 1), which may warrant further investigation.

To assess the tissue specificity of our fingerprint, we searched for the same proteins in metabolically important adipose and liver tissues. Despite detecting 94 and 82% of muscle fingerprint proteins across each tissue, respectively, both adipose and liver were depleted for pQTL presence (*Figure 3C*), suggesting that regulation of our candidate protein abundance is somewhat specific to skeletal muscle. Finally, we queried our fingerprint for any biological pathways that could represent novel drivers of insulin resistance by performing KEGG pathway enrichment (*Figure 3C*). Both 'endocytosis: mmu04144' (clathrin light chain A, clathrin light chain C, epidermal growth factor receptor [EGFR] pathway substrate 15) and 'insulin signalling pathway; mmu04910' (phosphorylase kinase regulatory subunit beta) featured in the top 10, providing further supportive evidence for the biological relevance of our fingerprint in the context of insulin sensitivity.

Next, we utilised CMAP to convert our fingerprint into a list of small molecules and ligands that promote or oppose our muscle insulin resistance fingerprint. To test our assumption that pQTL filtering would improve our fingerprint, we also queried CMAP with a list of the top 150 most strongly negatively associated proteins independent of pQTL presence. Intriguingly, on average CMAP scores for compounds and ligands were significantly higher when captured using a pQTL-filtered fingerprint compared to the non-pQTL filtered group, supporting the utility of this method (*Figure 3—figure supplement 1*). Encouragingly, the two highest scoring compounds identified using our fingerprint were Broad Institute glycogen synthase kinase (GSK3) and EGFR inhibitors. Both of these kinases have been independently identified as drug targets that reverse insulin resistance (*Leng et al., 2010*; *Timmons et al., 2022*; *Fazakerley et al., 2023*). Many compounds listed in the CMAP database are proprietary Broad Institute inhibitors that are only identified by their Broad ID and cannot be easily procured for follow-up experiments. Therefore, we excluded all Broad Institute compounds from further analysis. After this filtering, 856 small molecules and 91 ligands generated gene expression signatures matching our query. As whole-body insulin sensitivity decreased with increased fingerprint protein abundance, we focused on molecules whose CMAP score suggested a reversal of our insulin resistance fingerprint (*Figure 3D*). Ranking these candidates based on CMAP score revealed a number of well-known potentiators of insulin sensitivity including the antioxidant resveratrol (*Timmers et al., 2011*), the diabetes medication metformin (*Knowler et al., 2002*), and the growth factor FGF21 (*Geng et al., 2020*). We also identified the antibiotic thiostrepton (*Bailly, 2022*), a documented proteasome inhibitor, consistent with our enrichment analysis which identified the proteasome as negatively contributing to insulin sensitivity.

## Cross-validation of thiostrepton by Prestwick library screen of GLUT4 translocation

To obtain independent validation of some of the candidates revealed from CMAP, we performed a screen for compounds that affect GLUT4 translocation to the cell surface in L6 myotubes expressing HA-tagged GLUT4 (GLUT4-HA-L6), a readout of insulin action that is defective in insulin resistance. For this we used our established high-sensitivity, high-throughput 96-well plate format screen that is amenable to physiological models of insulin resistance (*Stöckli et al., 2008*; *Govers et al., 2004*), combined with the Prestwick library of U.S. Food and Drug Administration (FDA)-approved drugs. In

total, 420 compounds were found across both platforms, and these consensus compounds captured a significant proportion of highly scoring CMAP compounds (*Figure 4—figure supplement 1A and B*).

We performed three separate screens (*Figure 4A*) to capture the different mechanisms by which compounds modulate glucose uptake: (1) insulin-independent activation of GLUT4 translocation to the plasma membrane (basal activators), (2) potentiation of submaximal insulin action (insulin sensitisers), and (3) rescue of palmitate-induced insulin resistance (insulin resistance reversers). We identified 22 basal agonists (*Figure 4B*), 7 insulin sensitisers (*Figure 4C*), and 16 insulin resistance reversers (*Figure 4D*). Five compounds both stimulated GLUT4 translocation and reversed insulin resistance, four were both basal agonists and insulin sensitisers while none met all three criteria (*Figure 4E*). Overall, we found that compounds that were identified by CMAP score (*Figure 3D*) performed better as both basal activators and as insulin resistance reversers than those that did not (*Figure 4—figure supplement 1C*).

To cross-reference our CMAP data with the Prestwick screen in an unbiased way, we constructed a scoring matrix to rank compounds found by both our CMAP query and in the Prestwick library. First, we z-scored the values for each category (basal agonists, insulin sensitisers, insulin resistance reversal, CMAP score). Next, we averaged the three in vitro assays z-scores and added it to the CMAP score. This overall score represents how each compound modulates GLUT4 translocation and potentially reverses our insulin resistance fingerprint, relative to the rest of the compound library. Using this final value, we ranked each compound and displayed the top 20 in a heat map (*Figure 5A*). Based on this metric, thiostrepton was identified as the highest-ranking compound and was selected for subsequent validation by further GLUT4 translocation (*Figure 5B*) and 2-deoxyglucose uptake (*Figure 5C*) experiments in GLUT4-HA-L6 myotubes. We observed a consistent reversal of insulin resistance across both assays.

Next, we assessed the efficacy of thiostrepton to reverse insulin resistance in diet-induced obese mice. We decided to study it in isolated muscles as this circumvents potentially confounding microbiome effects due to thiostrepton's antibiotic activity and allows direct interrogation of muscle insulin action. We selected two strains of inbred mice, C57BL/6J and BXH9/TyJ, based on our previous observations that these strains are particularly amenable to developing muscle insulin resistance following high-fat, high-sugar (Western diet [WD]) diet feeding (*Nelson et al., 2022*). Consistent with diet-induced perturbations in metabolic health, both C57BL/6J and BXH9 BXH9/TyJ mice fed a WD had increased body weight and adiposity, fasting hyperglycaemia, fasting hyperinsulinemia, glucose intolerance, and lower systemic insulin sensitivity (Matsuda Index) relative to chow-fed controls (*Figure 5—figure supplement 1A–E*). WD feeding also resulted in ~40% reduction in C57BL/6J soleus insulin-stimulated 2-deoxyglucose uptake, a 75% reduction in BXH9 extensor digitorum longus insulin-stimulated 2-deoxyglucose uptake and 65% reduction in BXH9 soleus insulin-stimulated 2-deoxyglucose uptake. Strikingly, 1 hr of thiostrepton treatment prior to insulin addition was sufficient to reverse 80% of WD-induced insulin resistance in C57Bl/6J EDL muscle and 50% in BXH9 EDL muscle but did not restore BXH9 soleus 2-deoxyglucose uptake (*Figure 5D*).

## Thiostrepton does not affect insulin signalling

Next, we attempted to identify the potential mechanisms by which thiostrepton relieved insulin resistance. Canonically, insulin-stimulated GLUT4 translocation is facilitated by a signalling cascade comprising PI3K/Akt and dysfunction in this pathway has been implicated in insulin resistance (*Cho et al., 2001*), although this is controversial (*James et al., 2021*). To assess insulin signalling, we treated control and palmitate-treated GLUT4-HA-L6 myotubes with either thiostrepton or vehicle for 1 hr prior to insulin stimulation. Unlike GLUT4 translocation or 2-deoxyglucose uptake, palmitate did not perturb proximal insulin signalling. We detected no effect of palmitate treatment or thiostrepton on the phosphorylation of Akt-T308, Akt-S473, or the Akt-substrates GSK3-S21/9 and PRAS40-T246 (*Figure 6A–E*). These findings are consistent with the view that insulin resistance occurs independently of canonical insulin signalling (*Hoehn et al., 2008*; *Hoy et al., 2009*) and suggests that thiostrepton is acting independently of signalling to reverse insulin resistance.

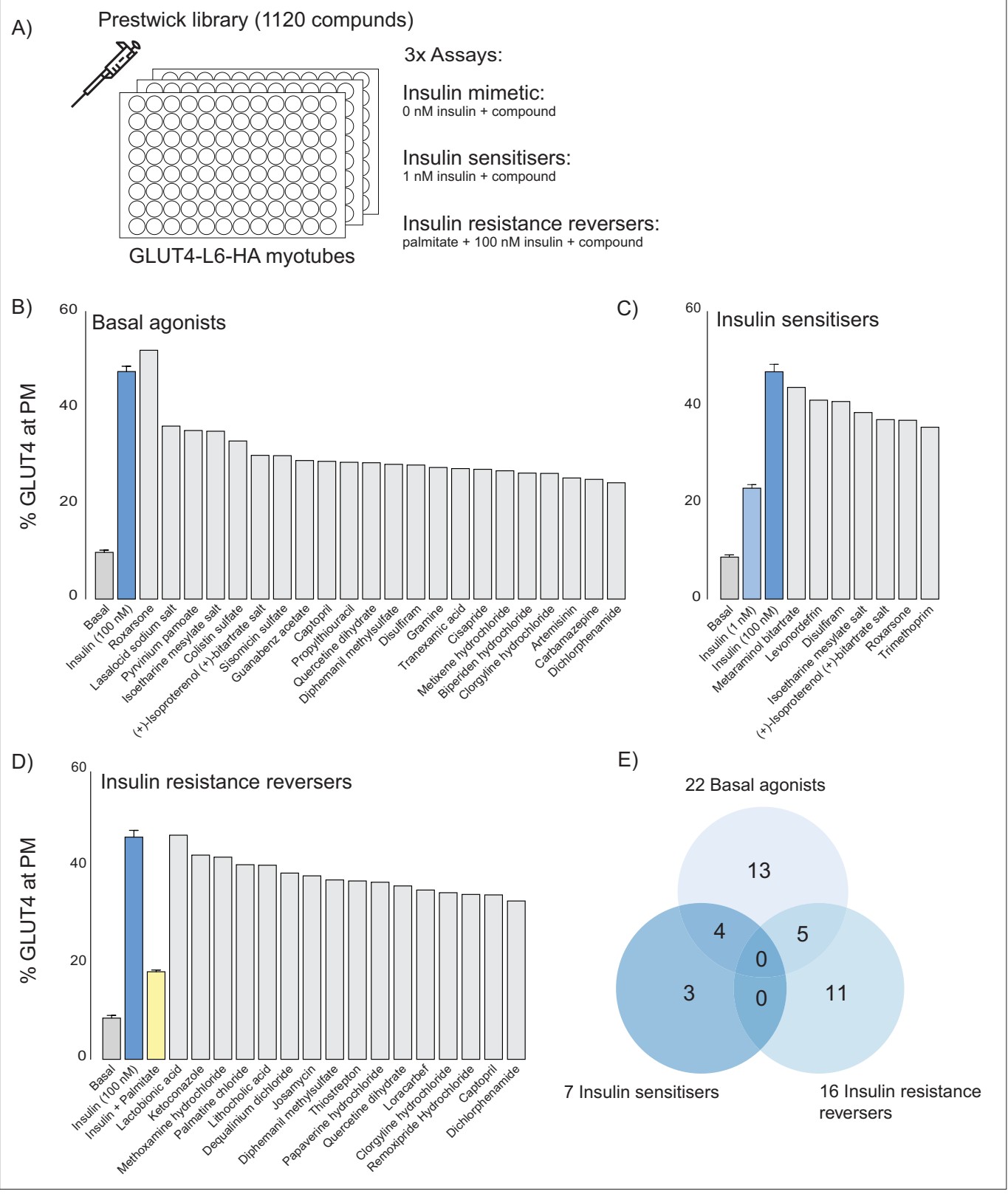

**Figure 4.** Prestwick library of U.S. Food and Drug Administration (FDA)-approved drugs that modulate GLUT4 translocation in L6 myotubes. (**A**) Schematic representation of the three assays performed. (**B**) Small molecules that promote GLUT4 exocytosis to the plasma membrane (PM) independently of insulin with controls (basal, insulin) on the left. (**C**) Small molecules that potentiate a submaximal dose of insulin (1 nM) with controls (basal, 1 nM and 100 nM insulin) on the left. Compounds were added in combination with 1 nM insulin. (**D**) Small molecules that reverse palmitate

*Figure 4 continued on next page*

*Figure 4 continued*

induced insulin resistance with controls (basal, insulin, insulin + palmitate) on the left. Compounds were added in combination with 100 nM insulin following palmitate treatment. (**E**) Venn diagram of compound overlap between assays. Biological significance for each assay was defined as 50% of corresponding control, see methods for details.

The online version of this article includes the following figure supplement(s) for figure 4:

**Figure supplement 1.** Validation of Connectivity Map (CMAP) results against Prestwick library of small molecules.

## Thiostrepton partially inhibits mitochondria and restores palmitate induced glycolysis suppression

Many insulin-sensitising agents act via mitochondrial inhibition or uncoupling (*Cameron et al., 2018*; *Madiraju et al., 2014*; *Alexopoulos et al., 2020*) and thiostrepton has been reported to inhibit mitochondrial translation (*Zhang et al., 2005*) and respiration (*Weinhaeuser et al., 2022*; *Cunniff et al., 2015*). To test whether thiostrepton's ability to restore optimal insulin action occurs via mitochondria, palmitate-treated and control GLUT4-HA-L6 myotubes were incubated with thiostrepton as above. Consistent with the mitochondrial dysfunction that has been reported during insulin resistance (*Anderson et al., 2009*; *Hoehn et al., 2009*), we observed substantial suppression of maximal mitochondrial respiration and mitochondrial reserve capacity following palmitate treatment (*Figure 7A*). Furthermore, thiostrepton alone appeared to blunt maximal respiration, albeit to a lesser extent than palmitate. When combined, thiostrepton and palmitate did not produce an additive suppression, nor did thiostrepton reverse any of the palmitate-induced defects. This suggests that this suppression of maximal respiration does not contribute to insulin resistance in this model.

We also assessed glycolysis by way of extracellular acidification rate (ECAR). Like mitochondrial respiration, palmitate suppressed maximal glycolytic capacity; however, unlike respiration this was potently reversed by co-treatment with thiostrepton (*Figure 7B*). To test whether this increase in glycolytic flux could be explained by changes in cellular energy status due to mitochondrial inhibition, we investigated the energy sensor AMP-dependent kinase (AMPK). AMPK can promote glycolysis (*Herzig and Shaw, 2018*), GLUT4 translocation (*Richter and Hargreaves, 2013*; *Jensen et al., 2008*), and glucose uptake in skeletal muscle independently of insulin. However, unlike the AMPK activator A-769662, we observed no effect of thiostrepton on the phosphorylation of AMPK or its substrate acetyl-CoA carboxylase in either control cells or cells treated overnight with palmitate (*Figure 7C–E*). Although A-769662 potently increases AMPK substrate phosphorylation in muscle cells, AMPK phosphorylation itself is not observed, consistent with a previous study (*Göransson et al., 2007*). These data suggest that if thiostrepton activates glycolysis via mitochondrial inhibition, it occurs independently of AMPK.

## Discussion

By leveraging genetic and phenotypic diversity of DOz mice, we have explored skeletal muscle's contribution to whole-body insulin action at the molecular level. Our approach was validated by the identification of various 'positive controls' at each level of analysis. Firstly, utilising adiposity as a covariate during linear modelling uncovered relationships between whole-body insulin sensitivity and muscle GLUT4 trafficking proteins; secondly, pathway enrichment revealed proteostasis (*Díaz-Ruiz et al., 2015*; *Guo et al., 2022*) and endocytosis (*Antonescu et al., 2008*; *Hall et al., 2020*) as key contributors to whole-body insulin sensitivity; and thirdly, querying CMAP with our fingerprint of insulin resistance returned metformin, GSK3 (*Leng et al., 2010*; *Lee and Kim, 2007*) and EGFR (*Timmons et al., 2022*) inhibitors as potential insulin resistance therapeutics (*Knowler et al., 2002*). The identification of these proteins, pathways, and drugs by our strategy gives us confidence in our approach and the novel players identified.

Systems-based approaches often identify networks as being drivers of disease. Empirical validation of these is difficult due to the complex interactions in biological systems. Here we took a drug-centric approach to validate our findings; this allowed targeting of entire pathways rather than singular nodes. We identified several compounds across both in silico and in vitro analyses which may restore muscle insulin action; indeed, several of these have previously been investigated. Disulfiram, sold under the brand name Antabuse, is used as an alcohol-dependency medication. Two studies have described

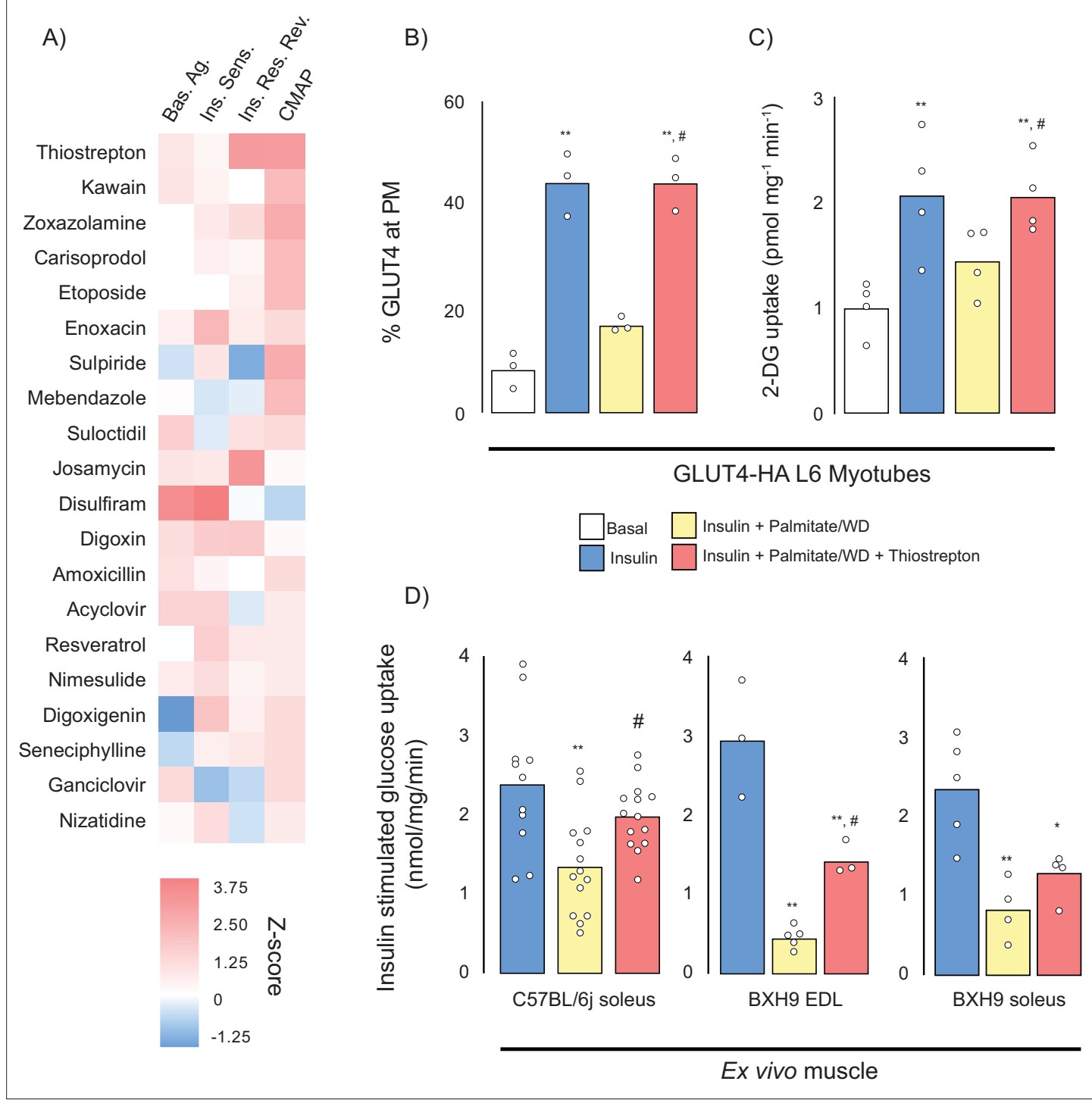

**Figure 5.** Cross-validation of Connectivity Map (CMAP) and Prestwick library. (**A**) Scoring matrix of top 20 scoring compounds present in both CMAP and the Prestwick library screens (basal agonists, Bas.Ag.; insulin sensitiser, Ins. Sens; insulin resistance reversers, Ins. Res. Rev). (**B, C**) Insulin stimulated GLUT4 translocation to the plasma membrane (PM) (**B**) and 2-deoxyglucose uptake (**C**) in control and insulin resistant L6 myotubes (palmitate) treated with thiostrepton or vehicle control. (**D**) Insulin-stimulated 2-deoxyglucose uptake in soleus and extensor digitorum longus (EDL) muscles from chow and Western diet (WD)-fed C57BL/6J and BXH9/TyJ following ex vivo treatment with thiostrepton or vehicle control. Data are mean with individual data points shown, n = 3–4 (**B, C**), n = 3–5 (BXH9); n = 11–14 (C57BL/6J). Significance was determined by one-way ANOVA with Student's post hoc test. ** indicates significant difference from control (basal or chow-fed C57BL6/J) group (p<0.01), * indicates significant difference from control (p<0.05). # indicates significant difference from palmitate-treated or WD-fed control group (p<0.05).

The online version of this article includes the following figure supplement(s) for figure 5:

**Figure supplement 1.** Effect of Western diet (WD) feeding on C57BL/6J and BxH9/TyJ mice body composition and insulin sensitivity.

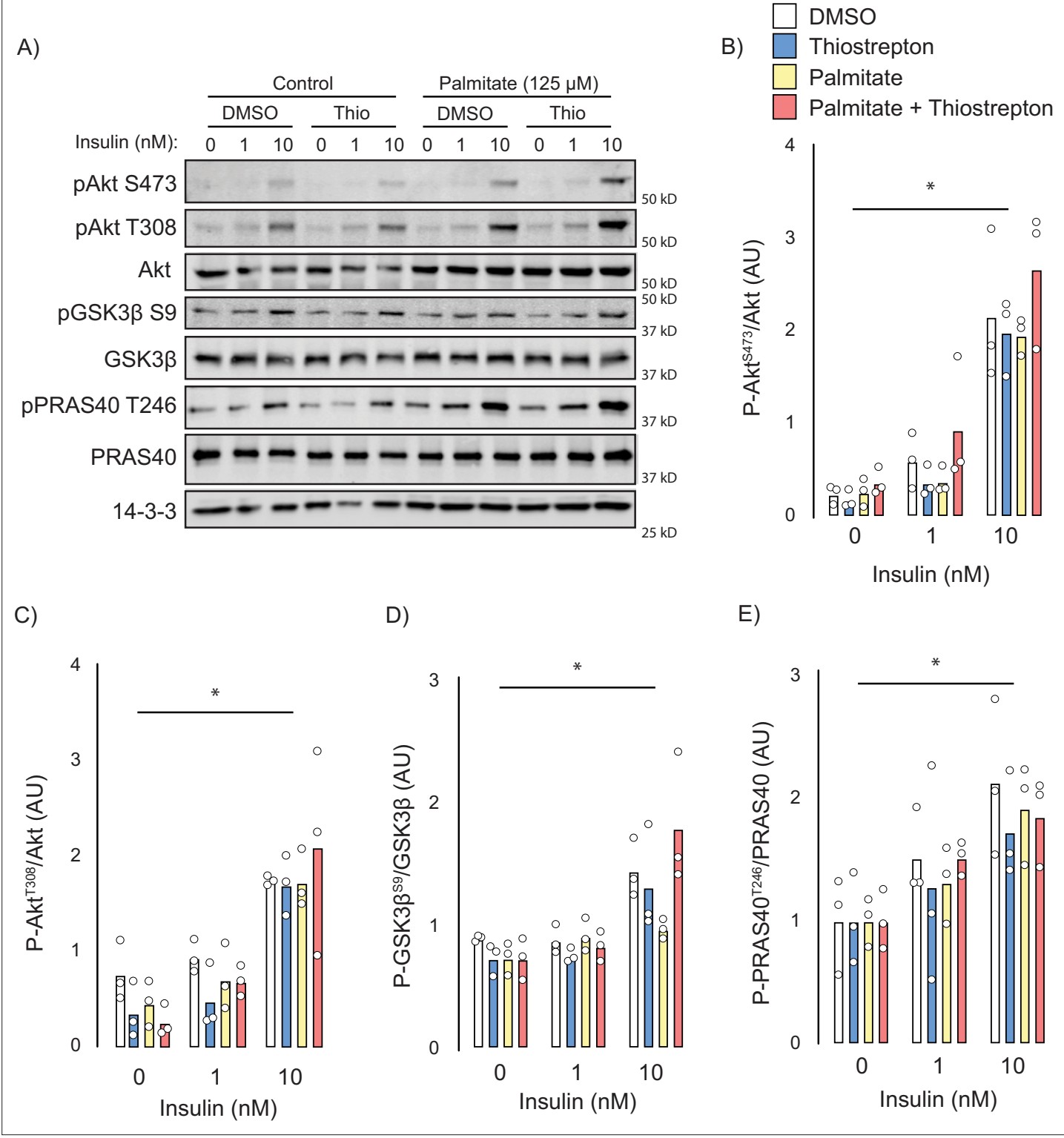

**Figure 6.** Effect of thiostrepton on insulin signalling. (**A–E**) Immunoblotting with indicated antibodies of control and palmitate treated GLUT4-HA-L6 myotubes following treatment with thiostrepton or vehicle control and stimulation with 0, 1, or 10 nM insulin. (**A**) Representative immunoblot shown of three independent experiments. (**B–E**) Quantification of immunoblots in (**A**) Akt S473 (**B**), Akt T308 (**C**), GSK3β S9 (**D**), and PRAS40 T246 (**E**). Data are mean with individual data points, n = 3. Significance was determined by two-way ANOVA with Student's post hoc test. * indicates a significant effect of insulin p<0.01.

The online version of this article includes the following source data for figure 6:

*Figure 6 continued on next page*

*Figure 6 continued*

**Source data 1.** Uncropped immunoblots with indicated antibodies of control and palmitate treated GLUT4-HA-L6 myotubes following treatment with thiostrepton or vehicle control and stimulation with 0, 1, or 10 nM insulin.

disulfiram's ability to reverse diet-induced hepatic insulin resistance and reduce adiposity (*Bernier et al., 2020a*; *Bernier et al., 2020b*). Resveratrol, a component found in red wine, is a popular antioxidant and has been demonstrated to reverse insulin resistance via reduction of reactive oxygen species (*Timmers et al., 2011*; *Gong et al., 2020*; *Shu et al., 2020*). Fibroblast growth factor 21 (FGF21) was also identified amongst the ligand dataset as reversing our insulin resistance fingerprint. FGF21 has previously been reported to promote insulin-stimulated glucose uptake in muscle fibres (*Rosales-Soto et al., 2020*) and can modulate mitophagy and proteostasis in muscle (*Oost et al., 2019*). Antivirals,

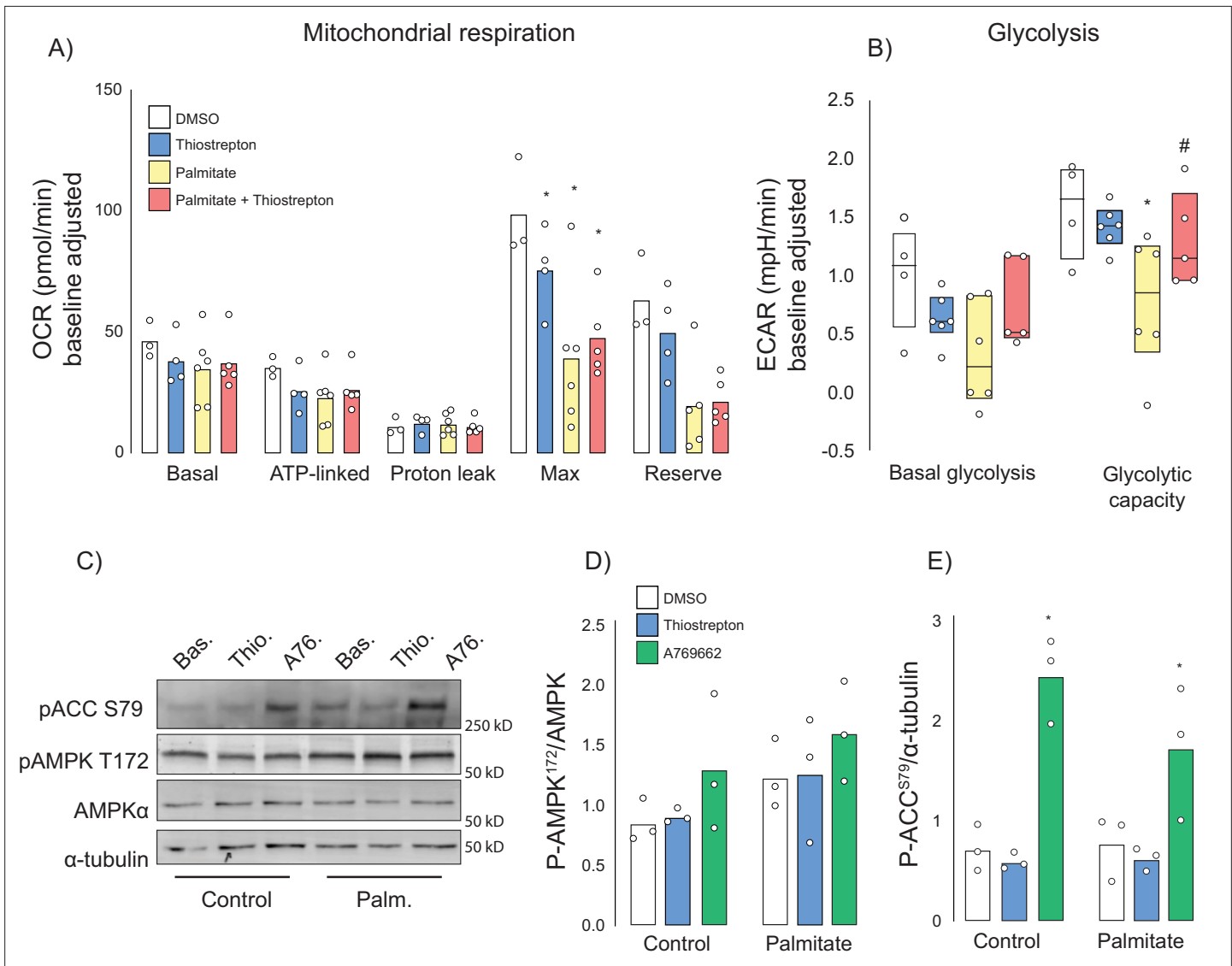

**Figure 7.** Effect of thiostrepton on mitochondrial respiration and glycolysis. (**A, B**) Oxygen consumption rates (**A**) and extracellular acidification rates (**B**) in control and palmitate treated GLUT4-HA-L6 myotubes treated with either thiostrepton or vehicle control. (**C**) Immunoblotting of AMPK signalling in control and palmitate-treated (Palm.) GLUT4-HA-L6 myotubes treated with either thiostrepton (Thio.), vehicle control (Bas.) or positive control A-769662 (A76). Representative immunoblots shown of three independent experiments. (**D, E**) Quantification of immunoblots in (**C**) AMPK T172 (**D**) and ACC S79 (**E**) phosphorylation following thiostrepton or A-769662 treatment. Data are mean with individual data points shown, n = 3. * indicates significant difference from control cells, # indicates significant difference from palmitate-treated cells p<0.05.

several classes of antibiotics, antipsychotics, and cancer drugs were all identified by our analyses. Perhaps this diversity reflects the divergent and pleiotropic biology of insulin resistance.

Key to our approach is the insulin resistance muscle proteomic fingerprint. This was generated by filtering proteins that associated with whole-body insulin sensitivity and that possessed significant *cis*-pQTLs. The latter was particularly important as we postulated that this would select for proteins that were likely to be causal drivers of insulin resistance. We hypothesised that a protein whose expression is post-translationally regulated in response to insulin resistance would not show a genetic signal and therefore be excluded from our fingerprint. Filtering on this basis improved the overall CMAP score and ultimately identified thiostrepton. One reason for this could be the discordance between mRNA and protein (*Maier et al., 2009*; *de Sousa Abreu et al., 2009*). CMAP uses mRNA expression data, whereas our fingerprint uses protein. By restricting our fingerprint to proteins with significant *cis*-pQTLs, we may have inadvertently selected for genes whose mRNA expression closely matches their protein, thereby increasing the overlap between fingerprint and perturbagen signatures. Moreover, our approach has the major advantage that it requires far fewer mice to obtain meaningful outcomes (222 mice in this study) compared to that required for genetic mapping of complex traits like Matsuda Index (*Gatti et al., 2014*). Furthermore, because we have used genetically diverse datasets (DOz mice and multiple cell lines in CMAP), our findings are likely robust across diverse target backgrounds.

A major question is what biological functions are represented by our fingerprint? One of the top pathways identified was endocytosis. This pathway featured two components of the clathrin coat and the adaptor protein EPS15. This is very exciting as endocytosis has been suggested to play a major role in stress signalling (*Cavalli et al., 2001*), and in the context of insulin sensitivity this may involve internalisation of key proteins including the insulin receptor and glucose or amino acid transporters (*Antonescu et al., 2008*; *Hall et al., 2020*). The concept that variation in this process is genetically determined, and this plays a major role in governing essential processes like insulin action, adds a new dimension to the role of this pathway. A second intriguing member of the fingerprint is phosphorylase kinase which, along with glycogen synthase kinase, regulates the key glycogen storage enzymes glycogen phosphorylase and glycogen synthase (*Thompson and Carlson, 2017*; *Polishchuk et al., 1995*; *Ding et al., 2000*; *Skurat et al., 2006*). High levels of glycogen phosphorylase kinase may promote glycogen breakdown through activation of glycogen phosphorylase, thereby altering GSK3 signalling, a process implicated in insulin resistance.

Neither AMPK nor Akt signalling account for the profound effect of thiostrepton on insulin action. This is exciting as it suggests both a novel mechanism of action and a novel insulin resistance defect. So far, the most enticing potential mechanism is restoration of glycolysis. Thiostrepton restores normal glycolytic function in palmitate-treated cells, and we have previously reported links between glycolysis and insulin action in skeletal muscle (*Nelson et al., 2022*; *Trefely et al., 2015*). Mechanistically, thiostrepton could promote glycolysis via attenuation of mitochondrial oxidative phosphorylation, and this has previously been demonstrated in acute myeloid leukaemia and malignant mesothelioma cell lines (*Weinhaeuser et al., 2022*; *Newick et al., 2012*). Our data supports this work and identifies a similar, albeit mild, effect on myotube respiration. Thiostrepton can also increase cellular exposure to mitochondrial reactive oxygen species (ROS) as it inhibits peroxiredoxin-3 (*Cunniff et al., 2015*; *Newick et al., 2012*), a key antioxidant enzyme. Perhaps, as seen during the Warburg effect (*Liu et al., 2015*; *Kulisz et al., 2002*; *Wang et al., 2020*), increased mitochondrial ROS can act as a signal to promote glycolysis and relieve mitochondrial energetic demands. Aside from glycolysis, other pathways may be involved in enhancing insulin sensitivity. For example, the negatively associated protein ARHGDIA (*Figure 2F*) is a potent negative regulator of insulin sensitivity, and our fingerprint of insulin resistance contained its homologue ARHGDIB. Both ARHGDIA and ARHGDIB have been reported to inhibit the insulin-action regulator RAC1 (*Liu et al., 2021*; *Gee et al., 2013*; *Sylow et al., 2013*), and thus may lower GLUT4 translocation and glucose uptake. Further investigations may uncover a role for thiostrepton in modulating the RAC1 signalling pathway via ARHGDIB.

Integration of physiological, proteomic, genomic, and pharmaceutical data has uncovered a potent reverser of insulin resistance. By integrating proteomic diversity with the underlying genetic architecture, we believe we were able to focus on potentially causal proteins, and the use of CMAP allowed us to combine these proteins into a single fingerprint to find potential modulators of insulin resistance. Our findings also build on recent reports linking glycolysis to insulin action and uncover a number of potential contributors to insulin action worthy of future study.

## Methods
### Mouse breeding and phenotyping

Male DOz mice (*Mus musculus*) were bred and housed at the Charles Perkins Centre, University of Sydney, NSW, Australia. They were originally established at Geniad, Western Australia, Australia, and then relocated to the University of Sydney. The DOz population comprises 46 breeding pairs and the breeding strategy avoids mating's between siblings or first cousins. Breeders are selected based on the genotype of the *R2d2* locus to limit the meiotic drive favouring the WSB allele on chromosome 2 (*Chesler et al., 2016*). The DOz mice used in the current study were outbred for 27–33 generations and comprised a total of 250 male DOz mice that were studied as 5 separate cohorts. Genomic DNA was isolated from each mouse and subjected to SNP genotyping (*Morgan et al., 2015*), followed by geno-typing diagnostics and cleaning as described (*Broman et al., 2019b*). Experiments were performed in accordance with NHMRC guidelines and under approval of the University of Sydney Animal Ethics Committee, approval numbers #1274 and #1988. To delineate genetic from cage effects, mice were randomised into cages of 3–5 at weaning. All mice were maintained at 23°C on a 12 hr light/dark cycle (0600–1800) and given ad libitum access to a standard laboratory chow diet containing 16% calories from fat, 61% calories from carbohydrates, and 23% calories from protein or a high-fat high-sugar diet (WD) containing 45% calories from fat, 36% calories from carbohydrate, and 19% calories from protein (3.5% g cellulose, 4.5% g bran, 13% g cornstarch, 21% g sucrose, 16.5% g casein, 3.4% g gelatine, 2.6% g safflower oil, 18.6% g lard, 1.2% g AIN-93 vitamin mix [MP Biomedicals], 4.95% g AIN-93 mineral mix [MP Biomedicals], 0.36% g choline and 0.3% g L-cysteine). Fat and lean mass measures were acquired via EchoMRI-900 (EchoMRI Corporation Pte Ltd, Singapore) at 14 wk of age. Glucose tolerance was determined by GTT at 14 wk of age by fasting mice for 6 hr (0700–1300 hr) before oral gavage of 20% glucose solution in water at 2 mg/kg lean mass. Blood glucose concentrations was measured directly by handheld glucometer (Accu-Chek, Roche Diabetes Care, NSW, Australia) from tail blood 0, 15, 30, 45, 60, and 90 min after oral gavage of glucose. Blood insulin levels at the 0 and 15 min timepoints were measured by mouse insulin ELISA Crystal Chem USA (Elk Grove Village, IL) according to the manufacturer's instructions. Blood glucose and insulin levels were integrated into a surrogate measure of whole-body insulin sensitivity using the Matsuda Index:

$$MatsudaIndex = \frac{10,000}{\sqrt{(Glucose_0 \times Insulin_0) \times (Glucose_{GTTmean} \times Insulin_{GTTmean})}}$$

### Muscle proteomic sample preparation

Whole quadriceps muscle samples were prepared as previously described with modification (*Frezza et al., 2007*; *Acin-Perez et al., 2020*). First, tissue was snap frozen with liquid nitrogen and pulverised before resuspension in 100 μl of trypsin buffer (phosphate-buffered saline [PBS] containing 10 mM EDTA and 0.01 ug/ul mass-spectrometry grade trypsin). Samples were incubated for 30 min at 37°C before being pelleted by centrifugation (10,000 × *g*, 5 min at 4°C). Samples were then resuspended in 1.4 ml mitochondrial isolation buffer (70 mM sucrose, 220 mM mannitol, 1 mM EGTA, 2 mM HEPES. pH at 7.4) and homogenised on ice in a glass Dounce homogeniser. Samples were then twice pelleted by centrifugation, first at 1000 × *g* × 10 min to remove insoluble debris and second at 10,000 × *g* × 10 min to extract the crude mitochondrial fraction, both centrifugation steps were performed at 4°C and the supernatant of the second step was collected as the post-mitochondrial fraction (PMF). The mitochondrial pellet was re-solubilised in 1 ml of isolation buffer by repeated pipetting on ice prior to centrifugation (10.000 × *g* × 10 min at 4°C) and resuspension in 50 μl of isolation buffer. Protein concentration of both mitochondrial and PMF was determined by BCA assay, 10 μg of protein was then prepared as previously described (*Nelson et al., 2022*). Reduction/alkylation (10 mM *tris* 2-carboxyethyl phosphine [TCEP], 40 mM CAA) buffer was added to each sample before incubation for 20 min at 60°C. Once cooled to room temperature, 0.4 mg trypsin and 0.4 mg LysC was added to each sample and incubated overnight (18 hr) at 37°C with gentle agitation. 30 μl water and 50 μl 1% TFA in ethyl acetate were added to stop digestion and dissolve any precipitated SDC. Samples were prepared for mass spectrometry analysis by StageTips clean up using SDB-RPS solid-phase extraction material (*Rappsilber et al., 2007*). Briefly, two layers of SDB-RPS material were packed into 200 μl tips and washed by centrifugation at 1000 × *g* for

2 min with 50 µl acetonitrile followed by 0.2% TFA in 30% methanol and then 0.2% TFA in water. 50 µl of samples were loaded to StageTips by centrifugation at 1000 × *g* for 3 min. Stage tips were washed with subsequent spins at 1000 × *g* for 3 min with 50 µl 1% TFA in ethyl acetate, then 1% TFA in isopropanol, and 0.2% TFA in 5% ACN. Samples were eluted by addition of 60 µl 60% ACN with 5% $NH_4OH_4$. Samples were dried by vacuum centrifugation and reconstituted in 30 µl 0.1% TFA in 2% ACN.

## Mass spectrometry analysis

Proteomic sample analysis was conducted using a Dionex UltiMate 3000 RSLCnano LC coupled to an Exploris Orbitrap mass spectrometer. Then, 2 µl of sample was injected on to an in-house packed 150 µm × 15 cm column (1.9 mm particle size, ReproSilPurC18-AQ) and separated using a gradient elution and with column temperature of 60°C, with Buffer A consisting of 0.1% formic acid in water and Buffer B consisting of 0.1% formic acid in 80% ACN. Samples were loaded to the column at a flow rate 3 µl/min at 3% B for 3 min, before dropping to 1.2 µL/min over 1 min for the gradient elution. The gradient was increased to 32% B over 50 min, then to 60% B over 0.5 min and 98% B over 0.5 min and held for 1.5 min, before returning to a flow rate of 3 µl/min at 3% B. Eluting peptides were ionised by electrospray with a spray voltage of 2.3 kV and a transfer capillary temperature of 300°C. Mass spectra were collected using a DIA method with varying isolation width windows (widths of *m/z* 22–589) between 350–1650 according to *Supplementary file 1*. MS1 spectra were collected between *m/z* 350 and 1650 at a resolution of 60,000 and an AGC target of 4e5 with a 50 ms maximum injection time. Ions were fragmented with stepped HCD collision energy at 27.5% and MS2 spectra collected between *m/z* 300 and 2000 at resolution of 30,000, with an AGC target of 3e5 and the maximum injection time of 54 ms.

Proteomics raw data files were searched using DIA-NN using a library-free FASTA search against the reviewed UniProt mouse proteome (downloaded May 2020) with deep learning enabled (*Rappsilber et al., 2007*; *Demichev et al., 2020*). The protease was set to Trypsin/P with one missed cleavage, N-term M excision, carbamidomethylation, and M oxidation options on. Peptide length was set to 7–30, precursor range 350–1650, and fragment range 300–2000, and false discovery rate (FDR) set to 1%. Both the PMF and mitochondrial fractions were filtered for mitochondrial proteins using based on MitoCarta 3.0 and presence in 50% in mice. Across both fractions we quantified 2073 proteins (1629 proteins in the PMF and 444 in the mitochondrial fraction). Proteomic intensities were log2 transformed and median normalised prior to analysis to achieve normal distributions and account for technical variation in total protein. The mass spectrometry proteomics data have been deposited to the ProteomeXchange Consortium via the PRIDE (*Perez-Riverol et al., 2022*) partner repository with the dataset identifier PXD042277.

## Data analysis

All data analysis and visualisation were performed in either the R programming environment (*R Development Core Team, 2013*) or GraphPad Prism (GraphPad Software, San Diego, CA). For protein-trait analysis, the Matsuda Index was calculated using glucose tolerance data before being log2 transformed. Linear models were generated using the *lm*() function in R where Matsuda Index = $\alpha$ + proteinX + covariate + $\varepsilon$ ($\alpha$ = intercept and $\varepsilon$ = error) using a Gaussian distribution (*Lehallier et al., 2019*). To correct for multiple testing, p-values were adjusted using the q-value method in the R package *qvalue* (*Dabney and Storey, 2010*). Chi-square tests for distribution differences within the data and two-/one-way ANOVA tests for group differences were performed in GraphPad Prism.

## Gene set enrichment

Gene set enrichment analysis for each mitochondrial and post-mitochondrial fraction was conducted using Matsuda Index effect sizes for each protein and performed in R using the *gseGO*() function within the clusterprofiler package (*Yu et al., 2012*). Over-representation analysis of protein–protein interaction clusters and KEGG pathway analysis of insulin resistance fingerprint proteins were performed in WebGestalt (*Liao et al., 2019*). All enrichment tests were performed using all quantified proteins within a given fraction as a background gene set. p-value correction was performed using FDR correction.

## Genetic mapping analysis

Genetic mapping analysis was performed in R using the QTL2 package (*Broman et al., 2019a*). The GIGA-MUGA single-nucleotide polymorphism array was used as genomic inputs for mapping (*Morgan et al., 2015*). pQTL analysis was performed by linear mixed modelling on *z*-scored protein abundance data with probabilistic estimation of expression residuals (PEER) factor adjustment, a covariate, and a kinship matrix to account for genetic relatedness amongst the DOz animals. PEER factor adjustment was performed using the top 10 calculated PEER factors, as described (*Stegle et al., 2012*). Significance thresholds were established by performing 1000 permutations and set at p<0.1 for *cis*-acting pQTL and p<0.05 for *trans*-acting pQTL. The *cis*-pQTL window was set as ±2 Mbp.

## CMAP and scoring matrix

Insulin resistance 'fingerprint' proteins were queried in CMAP using the CLUE software platform (*Subramanian et al., 2017*; *Lamb, 2007*). The list of 76 'fingerprint proteins' were queried against the L1000 gene expression dataset in the 'Query' function of CLUE, and results for small molecules (trt_cp) and ligands (trt_lig) were extracted using the CLUE 'Morpheus' platform. Raw connectivity score values were used to rank perturbagens. Connectivity scores were averaged across all cell lines in the L1000 dataset. The Broad Institute small-molecule inhibitors denoted by the prefix were removed from our resulting dataset as they are not commercially available. CMAP scores were combined with the results from our GLUT4 translocation screen to rank consensus compounds. This was done by first *z*-scoring each value (% GLUT4 at the plasma membrane for each assay and raw connectivity score). This produced a value which indicates how well each compound performs in a given test relative to the rest of the dataset. Then the average of all three GLUT4 assays (basal agonism, insulin sensitisation, and insulin resistance reversal) was added to the *z*-scored connectivity score to produce an overall score for each compound. *z*-score adjustment for each assay and CMAP score was performed as follows:

$$Zscore = \frac{x - mean}{SD}$$

where *Zscore* is the adjusted value for a given compound, *x* is the observed score for given compound, *mean* is the average score across all compounds, and *SD* is the standard deviation of all compounds.

Overall score for each compound was calculated as follows:

$$\frac{\text{z-scored } (bas) + \text{z-scored } (ins.sens) + \text{z-scored } (ins.res.rev)}{3} + \text{z-scored } (CMAP) = Overall\ score$$

## Cell culture

GLUT4-HA-L6 myoblasts (*Carey et al., 2006*) were grown in α-MEM supplemented with 10% fetal bovine serum. Differentiation was induced by changing media to α-MEM supplemented with 2% horse serum for 5 d.

## GLUT4 translocation assays

GLUT4 exocytosis was determined as previously reported (*Stöckli et al., 2008*). Briefly, GLUT4-HA-L6 myotubes were serum-starved overnight in α-MEM containing 0.2% BSA before being washed 3× with Krebs Ringer phosphate buffer supplemented with 0.2% BSA. Cells were stimulated with either 1 nM or 100 nM insulin for 20 min before being washed with ice-cold PBS and placed on ice. Cells were then fixed for 30 min in 3% paraformaldehyde and washed with PBS. The remaining paraformaldehyde was quenched with 50 mM glycine. Cells were then blocked for 20 min in either 5% normal swine serum (NSS) or 5% NSS with 0.1% saponin (for measurement of total GLUT4 levels). After blocking, cells were washed and incubated with anti-HA antibody (Convance, 1:200 in 5% NSS) for 45 min before incubation with secondary antibody for 20 min. Total and plasma membrane GLUT4-HA were determined by fluorescence plate reader at 485/520 nm. GLUT4 translocation was calculated as the percentage of total GLUT4 at the plasma membrane. For palmitate-induced insulin resistance assays, myotubes were incubated in α-MEM overnight supplemented with either 125 μM palmitic acid conjugated to BSA or equivalent BSA as a vehicle control before performing assay as above. For the assessment of

the Prestwick library of compounds, each compound was dissolved in DMSO and added for 1 hr at a final concentration of 10 μM (0.2% DMSO) prior to experimentation. Biological significance for each assay was defined as 50% of control. For basal agonists, this was >50% of 100 nM insulin; for insulin sensitisers, this was >50% of the difference between 1 and 100 nM insulin; and for insulin resistance reversers, this was >50% of the difference between 100 nM insulin and 100 nM insulin + palmitate.

## 2-Deoxyglucose uptake

2-Deoxyglucose uptake into GLUT4-HA-L6 cells was performed as previously described with modifications (*Carey et al., 2006*; *Masson et al., 2020*). Cells were incubated overnight in either α-MEM supplemented with either BSA-coupled 125 μM palmitic acid or BSA vehicle control before being washed 3× with HEPES buffered saline (HBS). Cells were then incubated in HBS supplemented with 10 μM unlabelled 2-deoxyglucose and 0.5 μCi/ml [$^3$H]-2-deoxyglucose at 37°C for 5 min. Cells were then washed 5× with ice-cold PBS and lysed in 1 M NaOH. For non-specific background uptake, one well per condition was pre-treated with cytochalasin B. Counts were determined by Perkin-Elmer Quantulus GCT Liquid Scintillation Counter (PerkinElmer, Waltham, MA). Glucose uptake was expressed relatively to protein concentration as determined by bicinchoninic acid (BCA) assay after subtraction of non-specific uptake.

## Ex vivo glucose uptake

Ex vivo glucose uptake was performed as previously described (*Nelson et al., 2022*). Mice were euthanised by cervical dislocation prior to rapid dissection of soleus (C57Bl/6J and BXH9/TyJ) and extensor digitorum longus (EDL) muscle (BXH9/TyJ only). Muscle selection was based on our prior observations that only soleus muscles in C57Bl/6J mice develop diet-induced insulin resistance. Both the soleus and EDL muscles were mounted and then incubated for 1 hr in Krebs Henseleit buffer (KHB; 5.5 mM glucose, 2 mM pyruvate, and 0.1% BSA) that had been gassed with carbogen (95% $O_2$/5% $CO_2$) supplemented with either 10 μM thiostrepton or a DMSO vehicle control. Glucose uptake was assessed by then switching the muscle into KHB supplemented with 0.375 mCi/ml [$^3$H]-2-deoxyglucose, 0.05 mCi/ml [$^{14}$C]-mannitol, 100 nM insulin, and either thiostrepton (10 μM) or DMSO vehicle control for 20 min at 30°C followed by washing in ice-cold PBS and then snap-frozen in liquid nitrogen. Samples were lysed in 250 mM NaOH at 70°C. Tracer in the muscle tissue lysates was quantified by liquid scintillation counting and [$^3$H]-2-deoxyglucose was corrected for extracellular [$^{14}$C]-mannitol then normalised to wet weight of the tissue.

## Immunoblotting

Glut4-HA-L6 myotubes were incubated overnight in 125 μM palmitate or BSA control prior to treatment with drugs/insulin as indicated. Cells were then washed in ice-cold PBS and lysed by scraping directly into 55°C Laemmli sample buffer with 10% TCEP. Samples were then sonicated for 24 s (3 s on/3 s off) and heated at 65°C for 5 min. SDS-PAGE was performed. Samples were resolved by SDS-PAGE as previously described (*Nelson et al., 2022*), transferred onto PVDF membranes and blocked in TBS-T (0.1% Tween in Tris-buffered saline) containing 5% skim milk for 1 hr. Membranes were then washed 3 × 10 min in TBS-T and incubated overnight in primary antibodies against phosphorylated Akt T308 (Cell Signaling Technologies #2965; diluted 1:1000), phosphorylated Akt S473 (Cell Signaling Technologies #9271; diluted 1:1000), pan-Akt (Cell Signaling Technologies #9272; diluted 1:1000), phosphorylated GSK-3α/β S21/9 (Cell Signaling Technologies #9327; diluted 1:1000), GSK3α/β (Cell Signaling Technologies #5676; diluted 1:1000), phosphorylated PRAS40 T246 (Cell Signaling Technologies #13175; diluted 1:1000), PRAS40 (Cell Signaling Technologies #2691; diluted 1:1000), phosphorylated AMPK (Cell Signaling Technologies, #2531; diluted 1:1000), α-tubulin (Cell Signalling Technologies #2125; diluted 1:1000), and 14-3-3 (Santa Cruz #sc-1657; diluted 1:5000). The following day membranes were washed 3 × 10 min in TBS-T and incubated for 1 hr in species-appropriate fluorescent or HRP secondary antibodies. Imaging and densitometry were performed using LI-COR Image Studio or a Bio-Rad ChemiDoc Imaging System (Bio-Rad, Hercules, CA) and ImageJ (*Schneider et al., 2012*). Phosphorylated proteins were normalised against their relevant controls, and data was normalised based on average band intensity.

## Mitochondrial stress test

Cellular respirometry (oxygen consumption rate [OCR]) was performed using Seahorse XFp miniplates and a Seahorse XF HS Mini Analyzer (Seahorse Bioscience, Copenhagen, Denmark) as previously described (*Yau et al., 2021*). GLUT4-HA-L6 myotubes were incubated overnight in either palmitate or BSA control αMEM before treatment with either thiostrepton (10 μM) or DMSO vehicle control. Cells were washed twice with KRBH and incubated in KRBH supplemented with 2.8 mM glucose, thiostrepton, or vehicle control without BSA (150 μl/well) at 37°C for 1 hr in non-CO₂ incubator. Cells were then assayed in XFp Analyzer. The OCR was measured after a 12 min equilibration period followed by 3/0/3 min of mix/wait/read cycles. Following stabilisation of baseline rates, compounds were injected sequentially to reach a final concentration of: 20 mM glucose, oligomycin (5 μg/ml), FCCP (1 μM), and rotenone/antimycin A (5 μM) to assess glucose-dependent respiration (calculated by baseline – glucose OCR), ATP-linked respiration (determined by glucose – oligomycin OCR), maximal respiration (calculated by FCCP – AntA/Rot OCR), and non-mitochondrial respiration, respectively (equal to AntA/Rot OCR). Data were normalised against protein concentration and presented as baseline adjusted.

## Acknowledgements

This work is dedicated to our wonderful colleague and friend Senthil Thillainadesan (1988–2023). We would like to thank the Sydney Mass Spectrometry facility in the Charles Perkins Centre at the University of Sydney for mass spectrometry support. We would also like to thank Large Animal Services in the Charles Perkins Centre at the University of Sydney for mouse housing support, and Cordula Hohnen-Behrens, Nicky Konstantopoulos, and Briana Spolding for in vitro GLUT4 translocation assay assistance.

## Additional information

### Competing interests

David E James: Senior editor, eLife. The other authors declare that no competing interests exist.

### Funding

| Funder | Grant reference number | Author |
| --- | --- | --- |
| Australian Research Council | Laureate Fellowship | David E James |

The funders had no role in study design, data collection and interpretation, or the decision to submit the work for publication.

### Author contributions

Stewart WC Masson, Conceptualization, Data curation, Formal analysis, Validation, Investigation, Visualization, Methodology, Writing – original draft, Project administration, Writing – review and editing; Søren Madsen, Conceptualization, Resources, Data curation, Software, Formal analysis, Investigation, Visualization, Methodology, Writing – original draft, Writing – review and editing; Kristen C Cooke, Data curation, Validation, Investigation, Methodology, Project administration, Writing – review and editing; Meg Potter, Validation, Investigation, Methodology, Project administration; Alexis Diaz Vegas, Data curation, Validation, Investigation, Methodology; Luke Carroll, Senthil Thillainadesan, Conceptualization, Software, Formal analysis, Investigation, Methodology; Harry B Cutler, Formal analysis, Visualization; Ken R Walder, Resources, Funding acquisition, Validation; Gregory J Cooney, Data curation, Investigation, Methodology; Grant Morahan, Resources; Jacqueline Stöckli, Conceptualization, Data curation, Validation, Investigation, Methodology, Writing – original draft, Project administration, Writing – review and editing; David E James, Conceptualization, Data curation, Funding acquisition, Investigation, Methodology, Writing – original draft, Project administration, Writing – review and editing

### Author ORCIDs

Stewart WC Masson http://orcid.org/0000-0003-4514-7009

Harry B Cutler (iD) http://orcid.org/0000-0002-2074-8599
Ken R Walder (iD) http://orcid.org/0000-0002-6758-4763
Gregory J Cooney (iD) http://orcid.org/0000-0003-0012-2529
David E James (iD) http://orcid.org/0000-0001-5946-5257

## Ethics

Experiments were performed in accordance with NHMRC guidelines and under approval of The University of Sydney Animal Ethics Committee, approval numbers #1274 and #1988.

Reviewer #1 (Public Review): https://doi.org/10.7554/eLife.86961.3.sa1
Reviewer #2 (Public Review): https://doi.org/10.7554/eLife.86961.3.sa2
Author Response https://doi.org/10.7554/eLife.86961.3.sa3

---

# Additional files

## Supplementary files

• Supplementary file 1. List of proteins and their Matsuda Index effect sizes which comprise our pQTL-filtered molecular fingerprint of insulin resistance.

• MDAR checklist

## Data availability

The mass spectrometry proteomics data have been deposited to the ProteomeXchange Consortium via the PRIDE partner repository with the dataset identifier PXD042277.

The following dataset was generated:

| Author(s) | Year | Dataset title | Dataset URL | Database and Identifier |
|---|---|---|---|---|
| Carroll L, Madsen S, Masson SWC, James DE | 2023 | Leveraging genetic diversity to identify small molecules that reverse mouse skeletal muscle insulin resistance | https://www.ebi.ac.uk/pride/archive/projects/PXD042277 | PRIDE, PXD042277 |

The following previously published datasets were used:

| Author(s) | Year | Dataset title | Dataset URL | Database and Identifier |
|---|---|---|---|---|
| Rath S, Sharma R, Gupta R, Ast T, Chan C, Durham TJ, Goodman RP, Grabarek Z, Haas ME, Hung WHW, Joshi PR, Jourdain AA, Kim SH, Kotrys AV, Lam SS, McCoy JG, Meisel JD, Miranda M, Panda A, Patgiri A, Rogers R, Sadre S, Shah H, Skinner OS, To TL, Walker MA, Wang H, Ward PS, Wengrod J, Yuan CC, Calvo SE, Mootha VK | 2021 | MitoCarta3.0: an updated mitochondrial proteome now with sub-organelle localization and pathway annotations | https://www.broadinstitute.org/mitocarta/mitocarta30-inventory-mammalian-mitochondrial-proteins-and-pathways | Broad Institute, mitocarta/mitocarta30-inventory-mammalian-mitochondrial-proteins-and-pathways |

*Continued on next page*

*Continued*

| Author(s) | Year | Dataset title | Dataset URL | Database and Identifier |
|---|---|---|---|---|
| Subramanian AN, Brooks AF, Vrcic AN, Flynn A, Rosains C, Takeda J, Hu DY, Davison R, Lamb D, Ardlie J, Hogstrom K, Greenside L, Gray P, Clemons NS, Silver PA, Wu S, Zhao X, Read-Button W, Wu W, Haggarty X, Ronco SJ, Boehm LV, Schreiber JS, Doench SL, Bittker JG, Root J A, Wong DE, Golub B, Todd R, Corsello R, Peck SM, Natoli DD, Gould X, Davis J, Tubelli JF, Asiedu AA, Lahr JK, Hirschman DL, Liu JE, Donahue Z, Julian M, Khan B, Wadden M, Smith D, Lam LC, Liberzon D, Toder A | 2017 | Connectivity Map (CMAP) | https://www.broadinstitute.org/connectivity-map-cmap | Broad Institute, connectivity-map-cmap |

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
