## [Editor Report · eLife assessment]

This **fundamental** study leverages natural genetic diversity in mice to discover candidate genes for insulin sensitivity, followed by experimental identification of compounds that can modulate insulin sensitivity, and finally initial mechanistic investigation of the mode of action. The generalized approach presented here, - the integration of systems genetics data with drug discovery -, supported by **compelling** evidence, will be a guide for others that seek to translate insights from mammalian genetics to drug discovery.

---

## [Referee Report · Reviewer #1 (Public Review)]

Masson et al. leveraged the natural genetic diversity presented in a large cohort of the Diversity Outbred in Australia (DOz) mice (n=215) to determine skeletal muscle proteins that were associated with insulin sensitivity. The hits were further filtered by pQTL analysis to construct a proteome fingerprint for insulin resistance. These proteins were then searched against Connectivity Map (CMAP) to identify compounds that could modulate insulin sensitivity. In parallel, many of these compounds were screened experimentally alongside other compounds in the Prestwick library to independently validate some of the compound hits. These two analyses were combined to score for compounds that would potentially reverse insulin resistance. Thiostrepton was identified as the top candidate, and its ability to reverse insulin resistance was validated using assays in L6 myotubes.

Below are several comments made on the original version of this study, addressed by the authors in the current version:

(1) Please describe the rationale of trypsinizing the tissue prior to mitochondrial isolation.

(2) The authors mentioned that the proteomics data were Log2 transformed and median-normalized. Please provide a bit more details on this, including whether the subjects were randomized.

(3) In Figure 1D, please give the numbers of mice the authors used for the CV comparisons in each group, whether they were of similar age and sex, and whether the differences in CV values were statistically significant

(4) The authors stated in lines 155-157 that proteins negatively associated with Matsuda index were further filtered by presence of their cis-pQTLs. Please provide more explanations to justify this filtering criterion.

(5) Please explain why the first half of the paper focused extensively on the authors' discoveries in the mitochondrial proteome, and how proteins involved in mitochondrial processes (such as complex I) were associated with Matsuda Index, but the final fingerprint list of insulin resistance, which contained 76 proteins, only had 7 mitochondrial proteins.

(6) The authors found that thiostrepton-induced insulin resistance reversal effects were not through insulin signalling. Please list the proteins in the fingerprint list that led to identification of thiostrepton on CMAP, and discuss whether you think that thiostrepton directly or indirectly acts on these protein targets.

---

## [Referee Report · Reviewer #2 (Public Review)]

In the present study, Masson et al. provide an elegant and profound demonstration of utilization of systems genetics data to fuel discovery of actionable therapeutics. The strengths of the study are many: generation of a novel skeletal muscle genetics proteomic dataset which is paired with measures of glucose metabolism in mice, systematic utilization of these data to yield potential therapeutic molecules which target insulin resistance, cross-referencing library screens from connectivity map with an independent validation platform for muscle glucose uptake and preclinical data supporting a new mechanism for thiostrepton in alleviating muscle insulin resistance. Future studies evaluating similar integrations of omics data from genetic diversity with compound screens, as well as detailed characterization of mechanisms such as thiostrepton on muscle fibers will further inform some remaining questions. In general, the thorough nature of this study not only provides strong support for the conclusions made but additionally offers a new framework for analysis of systems-based data. I had made several comments on the prior submission, all of which have been fully addressed and incorporated.

---

## [Author Response]

The following is the authors’ response to the original reviews.

**Reviewer #1 (Public Review):**
(1) What's the rationale of trypsinizing the tissue prior to mitochondrial isolation? This is not standard for subsequent proteomics analysis. This step will inevitably cause protein loss, especially for the post mitochondrial fractions (PMF). Treating samples with 0.01ug/uL trypsin for 37oC 30 min is sufficient to partially digest a substantial portion of the proteome. If samples from different subjects were not of the same weight, then this partial digestion step may introduce artificial variability as variable proportions of proteins from different subjects would be lost during this step. In addition, the mitochondrial protein enrichment in the mito fraction, despite statistically significant, does not look striking (Figure 1E, ~30% mitochondrial proteins in the mito fraction). As a comparison, Williams et al., MCP 2018 seem to have obtained high mitochondrial protein content in the mito fraction without trpsinizing the frozen quadriceps using a similar SWATH-MS-based approach.

Trypsinisation of the tissue prior to mitochondrial isolation is based on previous work and a Nature Protocol (1, 2) which isolated mitochondria for skeletal muscle. The rationale is that it aids in mechanical homogenisation from highly fibrous tissues such as quadriceps muscle by digesting extracellular matrix proteins. The trypsin/protein ratio used to aid in this process is at least 400 times lower than the amount of trypsin used for formal proteomic tryptic digestion. Three pieces of evidence suggest this step has negligible effect on downstream proteomic analysis. First, because the trypsinisation buffer is detergent free, trypsin will only affect extracellular or exposed membrane proteins. Filtering our PMF dataset for proteins with ‘extracellular matrix’ gene ontology identifies at least 90 unique extracellular matrix proteins indicating good retention of proteins susceptible to partial digestion. Second, the trypsin dose used is 50 times lower than the concentration used for passaging cultured cells, which retain viability after trypsinisation. Third, and contrary to the point raised by the reviewer, we observe less missingness in PMF samples compared to mitochondrial samples. We thank the reviewer for bringing the Williams et al. 2018 MCP paper to our attention. We note that mitochondrial enrichment between the two papers is comparable (~2- fold). To improve clarity line 408 now reads: “Whole quadriceps muscle samples were prepared as previously described with modification (99, 100). First, tissue was snap frozen with liquid nitrogen…” and line 95 reads: “Mitochondrial proteins were defined based on their presence in MitoCarta 3.0 (24) and consistent with previous work (25) were approximately two-fold enriched in the mitochondrial fraction relative to the PMF (Fig 1E).”

(2) The authors mentioned that the proteomics data were Log2 transformed and median- normalized. Would it be possible to provide a bit more details on this? Were the subjects randomized?

Samples were randomised prior to sample processing and mass spectrometry analysis. Because of possible variation in total protein content, it is critical to normalise protein intensities between samples. Median normalisation adjusts the samples so that they have the same median, thereby accounting for technical variation. Log2 normalisation helps to achieve normal distributions, critical for many downstream statistical tests. Line 471 now reads: “…to achieve normal distributions and account for technical variation in total protein.”

(3) In Figure 1D, what were the numbers of mice the authors used for the CV comparisons in each group? Were they of similar age and sex? Were the differences in CV values statistically significant?

The mitochondrial and PMF proteomes originated from the same quadriceps sample from the same mouse, and thus the age and sex are the same across both proteomes. After quality control, we had mitochondrial proteomes for 194 mice and PMF proteomes for 215 mice. The overall CV in the mitochondrial fraction was significantly greater than in the PMF, however whether the source of this variation is biological, or the result of mitochondrial isolation is unclear and as such we have avoided making a statement within the body of the manuscript. We have now more clearly described the nature of the samples in the revised manuscript and added sample sizes to figure 1F.

(4) The authors stated in lines 155-157 that proteins negatively associated with the Matsuda index were further filtered by presence of their cis-pQTLs. Perhaps more explanations would be needed to justify this filtering criterion? Having a cis-pQTL would mean the protein abundance variation is explained by the variation in its coding gene, this however conceptually would not be relevant to its association with the Matsuda index. With the data that the authors have in hand, would it not be natural to align the Matsuda index QTL with the pQTLs (cis and trans if available), and/or to perform mediation analysis to examine causal relationships with statistical significance?

The rationale for filtering by cis-pQTL was not to study the genetics of either Matsuda or associated proteins but rather to identify proteins that were more likely to be causally associated with Matsuda Index as opposed to adaptively associated. To clarify this line 165 now reads: “Filtering based on cis-pQTL presence was based on the rationale that if genetic variation can explain protein abundance differences between mice, then we can be confident that phenotype (Matsuda Index) is not driving the observed differences and therefore the protein-phenotype associations are likely causal. Importantly, this assumption can only be made for cis-acting pQTLs.” Previous work by Matthew et al. (see https://qtlviewer.jax.org/) has demonstrated that cis-pQTL have markedly higher LOD scores than trans-pQTLs, and our own unpublished work suggests that trans-pQTLs do not reproduce well between datasets. The reviewer rightfully suggests aligning protein QTL with those for Matsuda. This is our long-term goal but to identify genome wide significant peaks associated with altered Matsuda will require many more mice than studied here.

(5) It seems a bit odd that the first half of the paper focused extensively on the authors' discoveries in the mitochondrial proteome, and how proteins involved in mitochondrial processes (such as complex I) were associated with Matsuda Index, but the final fingerprint list of insulin resistance, which contained 76 proteins, only had 7 mitochondrial proteins. Was this because many mitochondrial proteins were filtered out due to no cis-pQTL presenting?

There are three reasons our fingerprint is lacking mitochondrial proteins: (1) there are more non-mitochondrial than mitochondrial proteins in the muscle proteome; (2) we focussed on negatively associated proteins, and as demonstrated in figure 2c, the mitochondrial proteome is enriched for positively associated proteins; (3) as implied by the reviewer, we filtered for pQTL presence, further reducing the number of mitochondrial proteins in our fingerprint. To improve clarity, line 170 now reads: “Low mitochondrial representation in the fingerprint is the result of selecting negatively associating proteins, and as seen (Figure 2C) previously, the mitochondrial proteome is enriched for positive contributors to insulin resistance.”

(6) The authors found that thiostrepton-induced insulin resistance reversal effects were not through insulin signalling. It activated glycolysis but the mechanism of action was not clear. What are the proteins in the fingerprint list that led to identification of thiostrepton on CMAP?Is thiostrepton able to bind or change the expression of these proteins? Since thiostrepton was identified by searching the insulin resistance fingerprint protein list against CMAP, it would be rational to think that it exerts the biological effects by directly or indirectly acting on these protein targets.

This is indeed the implication of our data. Because of the timescales involved it is unlikely that thiostrepton is changing fingerprint protein levels but could be binding to and inhibiting them. Searching the CMAP thiostrepton signature reveals ARHGDIB and NAGK as the fingerprint proteins with the most positive and negative fold-changes respectively perhaps suggesting they play a role in thiostrepton’s mechanism of action. Experiments are underway to test this hypothesis however these are beyond the scope of the current paper.

**Reviewer #2 (Public Review):**
Line 105: The observation that variance in respiratory proteins is stable while lipid pathways is variable is quite interesting. Is this due to lower overall levels of lipid metabolism enzymes (ex. do these differ substantially from similar pathways ranked from high-low abundance?).

The relationship between coefficient of variation (CV) and relative abundance of proteins is important to consider. To address this, we have now also performed GSEA on proteins ranked from high to low relative abundance. These comparisons have been added to supplementary figure 1 and line 110 now reads: “As a control experiment, we also performed enrichment analysis on proteins ranked by LFQ relative abundance. High CV pathways (enriched for high CV proteins) tended to be lower in relative abundance (enriched for low relative abundance proteins) (Supplementary Fig 1a, b). However, many high variability pathways, lipid metabolism for example, were not enriched in either direction based on relative abundance suggesting differences in relative abundance do not fully explain pathway variability differences.”

Line 154: the 664 associations are impressive and potentially informative. It would be valuable to know which of these co-map to the same locus - either to distinguish linkage in a 2mb window or identify any cis-proteins which directly exert effects in trans-

To assess this, we have analysed pQTL position relative to gene position to generate a ‘hotspot’ plot. We have also generated a histogram of this pQTL density (in a 2 Mbp window) and added these figures to figure 3. We did not detect any obvious pQTL hotspots, and the distribution of pQTLs across the genome appears fairly uniform. Line 159 now reads: “These were distributed across the genome and were predominately cis acting (Figure 3A)...”

Line 194: Cross-platform validation of the CMAP fingerprint results is an admirable set of validations. It might be good to know general parameters like how many compounds were shared/unique for each platform. Also the concordance between ranking scores for significant and shared compounds.

The Connectivity Map (CMap) query included 5163 compounds, the Prestwick library included 1120, and the overlap was 420. We have added these comparisons to supplementary figure 2. Supplementary figure 2 now also contains a comparison of CMap scores between overlapping compounds (found in CMap and the Prestwick library) against all significant compounds identified by CMap (supplementary figure 2b). Interestingly, compounds present in both platforms scored higher on average, suggesting the Prestwick library captures a significant proportion of highly scoring CMap candidates. Line 206 now reads: “In total, 420 compounds were found across both platforms, and these consensus compounds captured a significant proportion of highly scoring CMap compounds (Supplementary Figure 2A, B).”

Line 319: Another consideration in the molecular fingerprint is how unique these are for muscle. While studies evaluating gene expression have shown that many cis-eQTLs are shared across tissues, to my knowledge, this hasn't been performed systematically for pQTLs. Therefore, consider adding a point to the discussion pointing out that some of the proteins might be conserved pQTLs whereas others which would be more relevant here present unique druggable targets in muscle.

To examine tissue specificity, we determined whether our skeletal muscle fingerprint proteins were detected and contained a pQTL in two metabolically important tissues, liver and adipose. Despite detecting almost all the fingerprint proteins in both adipose and liver tissue, they were depleted for pQTL compared to skeletal muscle. These data have now been added to figure 3c. Line 172 now reads: “To assess the tissue specificity of our fingerprint we searched for the same proteins in metabolically important adipose and liver tissues. Despite detecting 94% and 82% of muscle fingerprint proteins across each tissue respectively, both adipose and liver were depleted for pQTL presence (Figure 3C) suggesting that regulation of our fingerprint protein abundance is specific to skeletal muscle.”

Line 332: These are fascinating observations. 1, that in general insulin signaling and ampk were not themselves shown as top-ranked enrichments with matsuda and that this was sufficient to alter glucose metabolism without changes in these pathways. While further characterization of this signaling mechanism is beyond the scope of this study, it would be good to speculate as to additional signaling pathways that are relevant beyond ROS (ex. CNYP2 and others)

We have now added further discussion to the manuscript to address this point., Line 347 now reads: “Aside from glycolysis, other pathways may be involved in enhancing insulin sensitivity. For example, the negatively associated protein ARHGDIA (Figure 2F) is a potent negative regulator of insulin sensitivity, and our fingerprint of insulin resistance contained its homologue ARHGDIB. Both ARHGDIA and ARHGDIB have been reported to inhibit the insulin action regulator RAC1 thus lowering GLUT4 translocation and glucose uptake. Further investigations may uncover a role for thiostrepton in modulating the RAC1 signalling pathway via ARHGDIB.”

Line: 314: Remove the statement: "While this approach is less powerful than QTL co- localisation for identifying causal drivers,", as I don't believe that this has been demonstrated. Clearly, the authors provide a sufficient framework to pinpoint causality and produce an actionable set of proteins.

We have edited line 314, which now reads: “Moreover, our approach has the major advantage that it requires far fewer mice to obtain meaningful outcomes (222 mice in this study) compared to that required for genetic mapping of complex traits like Matsuda Index.”

Line 346: I would highlight one more appeal of the approach adopted by the authors. Given that these compound libraries were prioritized from patterns of diverse genetics, these observations are inherently more-likely to operate robustly across target backgrounds.

This point is further supported by our thiostrepton results in both C57BL6/j and BXH9 mice. Line 317 now reads: “Furthermore, because we have used genetically diverse datasets (DOz mice and multiple cell lines in Connectivity Map) our findings are likely robust across diverse target backgrounds.”

Line 434: I might have missed but can't seem to find where the muscle data are available to researchers. Given the importance and novelty of these studies, it will be important to provide some way to access the proteomic data.

These data are now available via the ProteomeXchange Consortium. Line 465 now reads: “The mass spectrometry proteomics data have been deposited to the ProteomeXchange Consortium via the PRIDE (104) partner repository with the dataset identifier PXD042277.”

1. Frezza C, Cipolat S, Scorrano L. Organelle isolation: functional mitochondria from mouse liver, muscle and cultured filroblasts. Nat Protoc. 2007;2(2):287-95.

2. Acin-Perez R, Benador IY, Petcherski A, Veliova M, Benavides GA, Lagarrigue S, et al. A novel approach to measure mitochondrial respiration in frozen biological samples. The EMBO Journal. 2020;39(13):e104073.

3. Chick JM, Munger SC, Simecek P, Huttlin EL, Choi K, Gatti DM, et al. Defining the consequences of genetic variation on a proteome-wide scale. Nature. 2016;534(7608):500- 5.

4. Gatti DM, Svenson KL, Shabalin A, Wu L-Y, Valdar W, Simecek P, et al. Quantitative Trait Locus Mapping Methods for Diversity Outbred Mice. G3 Genes|Genomes|Genetics. 2014;4(9):1623-33.